# Metabolic Phenotypes in Asthmatic Adults: Relationship with Inflammatory and Clinical Phenotypes and Prognostic Implications

**DOI:** 10.3390/metabo11080534

**Published:** 2021-08-11

**Authors:** Adalberto Santos, Helena Pité, Cláudia Chaves-Loureiro, Sílvia M. Rocha, Luís Taborda-Barata

**Affiliations:** 1Faculty of Health Sciences, University of Beira Interior, Avenida Infante D. Henrique, 6200-506 Covilhã, Portugal; tabordabarata@fcsaude.ubi.pt; 2CICS-UBI, Health Sciences Research Centre, University of Beira Interior, 6201-506 Covilhã, Portugal; 3Medical Faculty, Agostinho Neto University, Luanda, Angola; 4Allergy Center, CUF Descobertas Hospital and CUF Tejo Hospital, 1350-352 Lisbon, Portugal; Helena.Pite@cuf.pt; 5Chronic Diseases Research Center (CEDOC), NOVA Medical School, Universidade Nova de Lisboa, 1150-082 Lisbon, Portugal; 6Pulmonology Unit, Hospitais da Universidade de Coimbra, Centro Hospitalar e Universitário de Coimbra, 3004-561 Coimbra, Portugal; cl_loureiro@hotmail.com; 7Faculty of Medicine, University of Coimbra, 3000-370 Coimbra, Portugal; 8Department of Chemistry & LAQV-REQUIMTE, University of Aveiro, Campus Universitário Santiago, 3810-168 Aveiro, Portugal; smrocha@ua.pt; 9Department of Immunoallergology, Cova da Beira University Hospital Centre, 6200-251 Covilhã, Portugal

**Keywords:** metabolomics, asthma, phenotypes, endotypes

## Abstract

Bronchial asthma is a chronic disease that affects individuals of all ages. It has a high prevalence and is associated with high morbidity and considerable levels of mortality. However, asthma is not a single disease, and multiple subtypes or phenotypes (clinical, inflammatory or combinations thereof) can be detected, namely in aggregated clusters. Most studies have characterised asthma phenotypes and clusters of phenotypes using mainly clinical and inflammatory parameters. These studies are important because they may have clinical and prognostic implications and may also help to tailor personalised treatment approaches. In addition, various metabolomics studies have helped to further define the metabolic features of asthma, using electronic noses or targeted and untargeted approaches. Besides discriminating between asthma and a healthy state, metabolomics can detect the metabolic signatures associated with some asthma subtypes, namely eosinophilic and non-eosinophilic phenotypes or the obese asthma phenotype, and this may prove very useful in point-of-care application. Furthermore, metabolomics also discriminates between asthma and other “phenotypes” of chronic obstructive airway diseases, such as chronic obstructive pulmonary disease (COPD) or Asthma–COPD Overlap (ACO). However, there are still various aspects that need to be more thoroughly investigated in the context of asthma phenotypes in adequately designed, homogeneous, multicentre studies, using adequate tools and integrating metabolomics into a multiple-level approach.

## 1. Introduction

Bronchial asthma is a chronic respiratory disease that affects individuals of all ages. It commonly involves chronic airway inflammation and symptoms of varying magnitude over time, which include dyspnea, chest tightness and cough [1]. It has a high prevalence, high morbidity and considerable levels of mortality [2]. According to the Global Initiative for Asthma (GINA), “Asthma is a heterogeneous disease, with different underlying disease processes. Recognizable clusters of demographic, clinical and/or pathophysiological characteristics are often called ‘asthma phenotypes’ [1]. In fact, multiple studies have shown that various subtypes of asthma can be reflected in external manifestations of the disease, which are designated as “phenotypes”, and may involve both clinical and inflammatory features, as well as others [3]. However, since asthma phenotypes do not imply any specific underlying pathophysiological mechanisms, asthma can also be classified into subtypes known as “endotypes” [4], which are based on specific pathophysiological mechanisms at both cellular and molecular levels [5,6,7].

The detection of biomarkers is necessary in order to obtain more robust definitions of phenotypes or endotypes of asthma [8,9,10]. This further helps to classify patients and may allow a better personalised therapeutic approach to each phenotype or endotype [11]. Although different types of biomarkers have been described, metabolic pathways also have components that are different between a healthy state and disease, and which may also be relevant as asthma biomarkers. Thus, the thorough analysis of small molecules such as amino acids, lipids, organic acids and nucleotides via metabolomics studies carried out on different biological samples—exhaled breath condensate (EBC), peripheral blood or urine—can be very important in the approach to asthma regarding diagnosis, monitoring, tailored treatment and prognosis, but many issues still need to be addressed. In fact, more specifically, metabolomics-associated biomarkers may be very useful for understanding asthma pathophysiology as well as various other aspects of the disease, including the prediction of exacerbation and response to treatment.

Metabolomics uses high-throughput analytic techniques which are combined with bioinformatics to obtain a thorough and detailed overview of multiple metabolites in biological sources, thereby being able to characterise healthy status- and disease-related metabolic signatures. Fast, targeted metabolomics and untargeted metabolomics are the two main study strategies in the area of metabolomics [12,13]. Both provide important information about changes in metabolism and quantification of metabolites in many chronic pathological settings, with applications in the diagnosis, pathophysiology and management of diseases, including asthma [14]. If, on the one hand, targeted metabolomics is only concerned with identifying and semi- or fully quantifying pre-defined metabolites of interest, the non-targeted strategy offers far more comprehensive results as to the identification and quantification of metabolites, since it does not restrict analysis to previously defined target molecules [12]. The latter is possibly the best way to characterise a disease from the metabolic point of view and identify new biomarkers [12,15]. However, the untargeted metabolomics strategy can be problematic because it identifies a wide range of metabolites that may be difficult to interpret and constitute a confounding factor. In fact, the identification and validation of relevant metabolites using untargeted metabolomics requires thoughtful analysis since only a subset of all metabolite features can be positively ascribed to a molecular structure [16,17]. Furthermore, a high level of computational analysis of big data is crucial for an adequate and standardised analysis and interpretation of results that may avoid or highly significantly minimise the possibility of yielding erroneous results [16,18,19,20]. This is very important because the metabolome can be influenced or confounded by many aspects such as age, sex, circadian rhythm, medication and other xenobiotics, microbiota, physical exercise, diet or even air pollution, both in healthy states and in disease. In addition, sample source and types, sample collection and storage aspects, analytical procedure aspects, as well as data analysis also influence results. Finally, external validation using results from different patient cohort samples is crucial to making results solid and generalisable; however, this aspect is lacking in many studies.

Methodologically, metabolomics strategies can be supported by several techniques, namely nuclear magnetic resonance “spectroscopy” (NMR) [21], liquid chromatography coupled to mass spectrometry (LC–MS) and gas chromatography coupled to mass spectrometry (GC–MS) [20,22]. LC–MS has possibly been the most used technique because it offers greater sensitivity in the identification of metabolites [15,23]. In any case, NMR is also an extremely useful technique, and the best and most thorough metabolomics approach would probably entail combining both techniques. In fact, jointly using LC–MS and NMR (LC–NMR–MS systems) allows the combination of high throughput (via NMR) with high sensitivity and resolution levels (via LC–MS) [24,25].

Electronic nose (eNose) devices can be used for global metabolite characterisation, by detecting complex mixtures of Volatile Organic Compounds (VOCs) in exhaled breath and providing associated breath-prints of such mixtures. eNose technologies are cheaper, non-invasive and provide more rapid capabilities, allowing for the earlier detection of metabolite changes compared to conventional analytical chemistry-based methods [26,27,28,29].

In the specific context of respiratory diseases, eNoses can detect changes in VOC mixtures in asthma [30,31,32], COPD [33,34,35], as well as in various other respiratory diseases, namely cystic fibrosis or tuberculosis [36,37,38,39]. Dual-technology eNoses are similar to conventional chemical identification approaches in having chemical analysis capabilities that allow them to identify VOCs as disease-specific biomarkers [30,31,32,33,34,35]. Finally, in this context, the use of application-specific database libraries of VOC biomarkers can favour early disease detection [29,40,41].

Overall, various metabolomics studies, focusing on small-molecule metabolites in urine, peripheral blood or EBC, including VOCs, have shown that metabolite expression can discriminate between (a) asthmatic and non-asthmatic individuals [32,42,43,44,45,46,47,48,49,50,51]; (b) asthmatic patients and patients with chronic obstructive pulmonary disease (COPD) [52]; (c) asthma exacerbations and stable asthma [53], (d) severe and non-severe asthma [54,55,56,57,58], (e) different asthma phenotypes [59,60,61] and (f) assessment of treatment responses and effects, including responsiveness or not to corticosteroids [13].

Most studies on biomarkers and phenotypes have mostly been performed in children and non-elderly adult asthmatics. In fact, phenotyping studies in elderly asthmatics are scarce and, to the best of our knowledge, no metabolomics approaches have been used in this subgroup of patients. This constitutes a major gap in knowledge because, in the past twenty years, there has been a clear increase in the percentage of the elderly population [62]. Furthermore, asthma is not always easy to diagnose or treat in such patients, due to multiple comorbidities, polypharmacy, partially different clinical manifestations, lower symptom awareness, failure to comply with medication or other problems [62,63]. Thus, having metabolomics biomarkers that may increase the diagnostic, prognostic and therapeutic capacity in personalised medicine approach becomes highly important in all age groups, particularly in the elderly.

This review will focus on some of the main aspects of metabolomics in adult asthma, particularly its use for the biomarker-based assessment of disease-characterising and differentiating features, namely in the study of possible metabolic biomarkers of different clinical or inflammatory asthma phenotypes and/or endotypes, discrimination between asthma and other chronic obstructive airway diseases in relation to inflammatory and clinical phenotypes and the eventual existence of actual metabolic phenotypes based on metabolomics signatures. The temporal stability and eventual prognostic capacity of metabolic signatures will also be briefly analysed, as well as other aspects that may influence such signatures. Results from studies in children will not be specifically mentioned since there have been two recent reviews focusing on this subpopulation and asthma [64] or atopic diseases, in general [65]. Since this is a narrative review, and not a systematic review, this document does not aim to be an exhaustive, comprehensive analysis but rather a conceptual approach to the issue of metabolomics signatures and asthma phenotypes.

## 2. Phenotypes and Endotypes in Adult Asthma

Several inflammatory and clinical phenotypes have been described in adult asthma, underlying the heterogeneity that can be observed in the disease. Inflammatory phenotypes are based on the main type of immune cells that are detected in association with the pathophysiology of disease, e.g., eosinophils or neutrophils [66,67], whereas clinical phenotypes are characterised according to symptoms, related diseases or responsiveness to treatment [66].

“Phenotypes” are external manifestations of asthma that result from the combination of hereditary and environmental influences [3,7,68], and which may occur in varying combinations [5,69,70] and even concurrently [71]. Furthermore, asthma phenotypes are not associated with a specific underlying pathophysiological mechanism, since different mechanisms may lead to the same phenotypic features, particularly in clinical terms [5,68,69,70,72].

The main inflammatory phenotypes that have been described in adult asthma, mostly, but not exclusively, involve eosinophilic, neutrophilic and paucigranulocytic types, and apply to severe asthma [69,73,74]. Furthermore, more recent phenotypic classifications have involved type-2 (T2) “high” inflammation, which is rich in T helper 2 (Th2) cells, in an IL-4, IL-5 and IL13-rich setting, and is frequently associated with eosinophilia [67,75], and T2 “low” inflammation, which is generally associated with IL-2, IFN-γ and/or IL-17 producing T cells and tends to be more frequently neutrophilic [67,76].

In addition, various phenotypes associated with disease features (onset of disease, functional and clinical aspects, asthma severity, response to treatment, among other aspects), and combinations thereof, have also been described. For instance, early-onset asthma (EOA) and late-onset asthma (LOA) are different onsets of disease-related “phenotypes”, with the former generally developing in childhood or early adulthood and the latter sometime in adulthood [77,78]. On the other hand, atopic asthma and non-atopic asthma are also regarded as different phenotypes, which are usually discriminated according to an association or not with sensitisation to aeroallergens [79,80]. In addition, severe asthma, steroid-resistant asthma, occupational asthma, aspirin-induced asthma, exercise-induced asthma and obesity-associated asthma have also been regarded as asthma phenotypes [69,79,81].

Finally, many studies have used cluster analysis for the detection of the aggregation of multilevel features that discriminate subgroups of asthma patients and which may be more relevant to real-world practice as well as in terms of prognostic significance. In this context, the most robust evidence obtained from studies of “aggregations” of asthma features involved the following phenotype “clusters”: (1) early-onset allergic asthma; (2) early-onset allergic moderate-to-severe asthma; (3) late-onset nonallergic eosinophilic asthma and (4) late-onset non-allergic non-eosinophilic asthma [82]. In addition, other biomarker studies in asthma, involving international networks and studies such as ADEPT and U-BIOPRED, found similar, but not completely superimposable, phenotype clusters: (1) “mild, good lung function, early onset” asthma, associated with a low degree of predominantly type-2 (T2) inflammation; (2) “moderate, hyper-responsive, eosinophilic” asthma, with moderate asthma control, mild airflow obstruction and predominant type-2 inflammation; (3) “mixed severity, predominantly fixed obstructive, non-eosinophilic and neutrophilic” asthma, with moderate asthma control and low type-2 inflammation; (4) “severe uncontrolled, severe reversible obstruction, mixed granulocytic” asthma, with a moderate degree of type-2 inflammation [83,84].

Several previously described asthma “phenotypes” have subsequently been proposed as “endotypes”, and examples of these include adult allergic asthma, aspirin-sensitive asthma, late-onset hypereosinophilic asthma, obesity-related asthma, neutrophilic asthma, T2-high or T2-low (non-Th2-high) asthma [5,85,86]. In fact, as is the case of T2-high and T2-low asthma, underlying immunopathophysiological mechanisms (involving cytokines) have been identified and there is even effective medication targeted at these mechanisms (e.g., anti-IgE, anti-IL-5, anti-IL-5 receptor or anti-IL4/IL3 receptor α-chain) [87]. Figure 1 shows examples of possible phenotypes and endotypes of asthma.

## 3. Main Metabolomic Signatures and Their Potential Implications in Adult Asthma

Besides the inflammatory aspects, metabolic changes are also an element that must be taken into account as a basis for the molecular understanding of the underlying pathophysiological processes in asthma. In fact, some metabolic pathways may be activated or changed otherwise and studying such changes may help us to understand the metabolic pathways that can be implicated in the pathogenic process, in association with inflammatory aspects. However, because metabolic processes are dynamic and reflect many influences, namely genome–environmental interactions and the effects of asthma and its severity on the metabolome, need to be distinguished from effects due to treatment and other confounders. This is all the more important in persistent asthma that requires regular inhaled corticosteroids (ICS), or in severe asthma, which may require short bursts of oral corticosteroids, and it may also apply to other types of medication both for asthma and for other concurrent diseases, as is the case in elderly individuals who frequently have multimorbidity.

A correct diagnosis and optimal treatment approach for each asthma patient remain a challenge for healthcare professionals. The heterogeneity of the disease, involving different pathophysiological mechanisms and/or disease expression features, as can be inferred from the various phenotypes and clusters of phenotypes, as well as endotypes, makes its correct approach complex. In view of these difficulties, metabolomics emerges as a strategy that allows the identification of altered metabolic pathways, as well as the metabolites eliminated from these pathways in asthma, thereby improving their identification as general biomarkers of the disease, and of asthma subtypes, response to treatment or even as novel therapeutic targets [88,89].

This review will only briefly comment on how well metabolomics can differentiate between asthmatic and non-asthmatic individuals, between asthma exacerbations and periods of stable asthma or between severe and non-severe asthma. It will rather concentrate on how metabolomics can discriminate between asthma and other subtypes of chronic obstructive airway diseases, as well as between different clinical and/or inflammatory asthma phenotypes and endotypes, including eosinophilic and non-eosinophilic asthma, obesity-associated asthma or steroid-resistant asthma. The issues of phenotype stability and the potential prognostic value of metabolomics-detected biomarkers in asthma will also be addressed. Overall, there is heterogeneity across studies, for various reasons, and some results have not been replicated or externally validated or data are simply scant or even non-existent. These aspects imply that this exciting and interesting field warrants further research.

Most metabolomics studies can detect asthmatic patients, although evidence is more robust in terms of differentiating between healthy controls and patients with severe asthma or patients with an exacerbation of the disease. Positive results have been described in exhaled breath, via the study of VOCs and using either eNoses or targeted or untargeted metabolomics analysis, in peripheral blood and in urine, using various NMR or GC approaches [30,31,32,42,43,44,45,46,50].

It is possible to derive certain notions concerning a metabolic asthma profile in comparison with that of healthy controls, in spite of clear heterogeneity across different studies. Such heterogeneity involves metabolomics analysis/detection methods (e.g., NMR versus LC–MS or GC–MS versus other forms or combined forms of analysis), studied metabolites (lipids versus amino acids, versus carbohydrates or multiple combinations thereof, as well as other metabolites) and also the types of asthmatic patients that were included (diagnosed according to different criteria, with different percentages of patients on inhaled treatment, or with different levels of disease severity, among other factors) [30,42,43,44,45,46,50,51,54,90,91]. Since the analysis of metabolomics differences between asthmatic individuals and healthy controls has been the focus of various reviews, we will only succinctly summarise some of the main findings.

Ethane levels in exhaled breath may be higher in asthma but this difference may be more robust or significant in steroid-free asthmatics [26,30]. In any case, elevated levels may thus be a noninvasive marker of oxidative stress in asthma.

In addition, metabolic pathways such as glycolysis and gluconeogenesis may also be altered in asthma, as suggested by the increased levels of carbohydrates such as glucose in the airway epithelium of asthmatics [91]. This is relevant because glucose, for example, is involved in the production of reactive oxygen species (ROS), which enables the activation of the inflammasome in asthmatics.

Regarding lipidomics, lipid metabolites such as phosphatidylethanolamine (PE) (18:1p/22:6), PE (22:0/18:1), PE (38:1), sphingomyelin (SM) (d18:1/18:1) and triglyceride (TG) (16:0/16:0/18:1) may be increased in more severe asthmatics, whereas metabolites such as phosphatidylinositol (PI) (16:0/20:4), TG (17:0/18:1/18:1), phosphatidylglycerol (PG) (44:0), ceramide (Cer) (d16:0/27:2) and lysophosphatidylcholine (LPC) (22:4) may be decreased [51]. Furthermore, PE (38:1) was the main lipid metabolite that better discriminated between asthmatic individuals and healthy controls. Finally, targeted lipid metabolomics showed that individuals with severe asthma have a higher content of phosphatidylcholines (PCs), LPCs, lysophosphatidylethanolamines (LPE) and bis (-monoacylglycer) phosphates in bronchial epithelial cells, compared with non-asthmatic individuals and those with mild and moderate asthma [90].

SM and cholesterol are of great importance in several functions, such as transmembrane signal transduction, which is involved in apoptosis, cell proliferation, differentiation, inflammation and oxidative stress. On the other hand, elevated levels of PE (particularly in oxidised forms) are indicative of programmed cell death via ferroptosis, since PE is an important phospholipid of the glycerophospholipid class that constitutes the plasma membrane (inner leaflet) of viable cells and is released when cells enter cell death [92].

Peripheral blood taurine, lathosterol, bile acids, nicotinamide and adenosine-5-phospate levels have also been shown to be significantly higher in asthmatics than in healthy controls [54].

In summary, metabolomics can discriminate between healthy individuals and patients with bronchial asthma. Globally, the most frequently identified metabolites, although with differences across studies, have involved energy homeostasis, lipid metabolism, tricarboxylic acid cycle, oxidative stress, hypoxia-associated molecules and various metabolites associated with immunoinflammatory processes that may be relevant to the underlying immunopathology of asthma [49,93,94]. Most of these affected metabolic pathways are concordant with the fact that asthma is a chronic, inflammatory disease, which requires changes in energy supply and possibly also metabolic reprogramming, namely in immune cells involved in the process.

However, it is important to ascertain whether metabolomics approaches can detect any differences between various clinical and/or inflammatory asthma phenotypes and/or endotypes, an aspect that may be crucial in providing non-invasive, point-of-care support for clinical decisions regarding individual patients.

Figure 2 shows the main altered metabolic pathways and the main metabolites observed in targeted and non-targeted metabolomics studies in asthma, as well as some general implications for the disease.

## 4. Assessment of Metabolic Changes in Inflammatory Asthma Phenotypes

Although asthma phenotypes can have some overlap, they may still have prognostic implications. Thus, it would be interesting to analyse whether they are also associated with different metabolic signatures—in other words, whether metabolomics can be used to detect inflammatory asthma phenotypes.

### 4.1. Global Metabolomic Signatures in Eosinophilic and Non-Eosinophilic Asthma Phenotypes

Various studies using eNoses [95,96,97] or other approaches [59] analysed VOCs in exhaled breath to classify asthma in adults into different inflammatory phenotypes. This is relevant in terms of point-of-care management of asthma patients, particularly those with more severe disease [98]. In addition, other authors [54,91] used untargeted metabolomics with the same purpose.

Plaza et al. [95] performed a metabolomics study of whether an eNose could differentiate between eosinophilic, neutrophilic and paucigranulocytic inflammatory phenotypes in 52 patients with persistent asthma. Such phenotypes were determined by cell counts in induced sputum. VOCs breath-prints were analysed using discriminant analysis on principal component reduction, which allowed calculation of cross-validated accuracy values. In addition, receiver operating characteristic (ROC) curves were calculated. Results showed that the eNose-derived breath-prints were different in eosinophilic asthmatics compared with neutrophilic as well as with paucigranulocytic asthma. Furthermore, the neutrophilic and paucigranulocytic breath-prints were also significantly different. Thus, again, an exhaled breath study of VOC could differentiate between inflammatory phenotypes in patients with persistent asthma.

Fens et al. [96] carried out a more invasive bronchoscopic study that aimed to ascertain whether an eNose could identify adult asthmatic patients with bronchoalveolar lavage fluid (BALF) eosinophilia, in order for this metabolomics approach to be used as a point-of care approach to differentiate between patients with eosinophilic and non-eosinophilic asthma. Thirteen patients with mild asthma (6 females; 7 males) were recruited and all were ex-smokers. Multiple regression analysis showed that the eNose breath-print was significantly associated with BALF eosinophilia but not with BALF neutrophils, macrophages or lymphocytes. The authors also performed exhaled breath nitric oxide (FeNO) analysis but no association was found with any inflammatory cell predominance. In fact, this parameter has shown less consistent results across different studies [99,100,101,102]. In any case, FeNO and eosinophilic inflammation in the bronchi may have at least partially distinct pathways and disease expression, although both can be markers of T2-high asthma phenotype/endotype [10,103].

Brinkman et al. [97] performed a multicentre analysis of exhaled metabolomics fingerprints from various eNoses in an adult U-BIOPRED cohort of severe asthma patients, in order to ascertain whether it was possible to identify different inflammatory phenotypes with such a metabolomics approach. The authors carried out unsupervised Ward clustering enhanced by similarity profile analysis in conjunction with K-means clustering. Various exploratory analyses were performed through partial least-square discriminant analysis (PLS-DA). Importantly, three eNose-detected metabolomics clusters became apparent, which discriminated between circulating and neutrophil percentages in peripheral blood. These metabolomics clusters were as follows: cluster 1 (33% of patients) did not show an evident unbalance towards preferential peripheral blood neutrophilia or eosinophilia; cluster 2 (42% of patients) had the highest mean percentage of peripheral blood neutrophils and included the highest percentage of patients on oral corticosteroids; cluster 3 (24% of patients) had the highest mean percentage of peripheral blood eosinophils and the lowest percentage of patients on oral corticosteroids. Eosinophilia and neutrophilia were also analysed in induced sputum but no significant differences were found among the clusters. Finally, although this study used three different clustering techniques that showed consistent results, it did not include an external validation cohort or even a training and a validation set within the same cohort.

A study by Ibrahim et al. [59], which adequately differentiated between healthy state and asthma, was not able to accurately discriminate between an eosinophilic asthma phenotype and non-eosinophilic phenotypes. This study analysed EBC metabolomics by NMR in 82 elderly and non-elderly adult asthmatic patients (76% atopic; mostly female) and in 35 healthy controls. Inflammatory phenotypes were defined according to induced sputum eosinophilia and neutrophilia, and multivariate modelling was performed on 70% of the total sample to obtain a discriminatory model that allowed discrimination between asthmatics and healthy controls, and the model was then tested in the remaining individuals. Further analyses were carried out to determine the most adequate models for the identification of metabolic signatures (in NMR spectral regions) in eosinophilic and neutrophilic inflammatory phenotypes as well as regarding asthma control and inhaled corticosteroid use. However, the approach had variable success in classifying asthma phenotypes, since it could discriminate patients with neutrophilic asthma but not those with eosinophilic asthma.

In a different study, Comhair et al. [54] performed an ultra-HPLC/tandem MS (MS/MS) untargeted as well as a focused analysis of plasma metabolomic profiles in 20 asthmatics (10 severe; 10 non severe, with severity defined in accordance with ATS Refractory Asthma Workshop criteria) [104] and 10 healthy controls. In brief, these criteria imply that, among other parameters, these patients were on regular or almost regular oral corticosteroids as well as on regular ICS. Patients were stratified according to asthma severity or by levels of fraction of exhaled nitric oxide (normal if FeNO < 35 parts per million (ppm) or high if ≥35 ppm). Nine patients had normal FeNO levels and nine had elevated levels. Besides discriminating between asthmatic patients and healthy individuals, this study also showed that more severe patients expressed changes in steroid and amino acid/protein metabolism in comparison with those with less severe asthma. In addition, patients with high levels of FeNO (possibly expressing T2-type bronchial inflammation) had higher plasma levels of branched chain amino acids, such as isoleucine, valine and 3-hydroxyisobutyrate, and bile acids such as glycocholate and cholate. In particular, this report by Comhair et al. suggested that severe asthmatics and those with high FeNO levels (possibly eosinophilic or T2) have metabolic signatures that indicate possible changes in NO-related taurine transport and bile acid metabolism.

Finally, Pang et al. [91] studied the relationship between some previously described inflammatory asthma phenotypes and metabolic features in mild asthmatic patients, diagnosed in accordance with the GINA 2016 guidelines. This study included 13 patients with eosinophilic asthma and 16 with non-eosinophilic asthma, who were classified into these phenotypes according to a score that included eosinophil/lymphocyte and eosinophil/neutrophil peripheral blood ratios. In addition, 15 healthy controls were also studied. Untargeted metabolomics analysis of peripheral blood involved ultra performance liquid chromatography–tandem mass spectrometry (UPLC–MS/MS). Although with some variability across different models based on principal component analysis (PCA), an orthogonal partial least squares data analysis (OPLS-DA) model showed that 18 different metabolites were differentially expressed in the three groups, reflecting changes in various metabolic pathways associated with immune regulation, energy and nutrients, the most relevant of which were related to the metabolism of glycerophospholipids, sphingolipids and retinol (which was decreased in asthmatics, particularly with the eosinophilic phenotype); these may eventually be used as biomarkers of these phenotypes. As examples, monosaccharides, PC (18:1/2:0), PS (18:0/20:0) and arachidonic acid (AA) showed higher levels in non-eosinophilic asthma, whereas PE (18:3/14:0), PC (16:0/18:1), LPC (18:1) and lactosylceramide (d18:1/12:0) levels were higher in eosinophilic asthma. These results are interesting but should be interpreted with caution since this study had a small-sized sample for performing robust metabolomics analyses, and studies with higher numbers of patients with different inflammatory phenotypes and controls are warranted.

In any case, low retinol levels in asthmatics may be associated with an inability to suppress eosinophil differentiation and eosinophilic inflammation in asthma, because retinol has anti-inflammatory and antioxidant properties, and helps to repair the airway epithelium [105]. Since retinol is a precursor of retinoic acid, it is interesting to observe the potential effects of these retinoids in inflammatory cell differentiation. In fact, an in vitro study showed that retinoic acid reduced IL-4-induced eotaxin expression in human bronchial epithelial cell line BEAS-2, suggesting that retinoic acid may reduce eosinophilic airway inflammation in diseases such as asthma. Furthermore, at least in the bone marrow of non-atopic individuals, retinoic acid inhibits IL-5 receptor expression on eosinophil–basophil precursors and the differentiation of these cells [106]. Thus, the observed decrease in retinol levels in eosinophilic asthma may at least partially contribute to the pathophysiology of this inflammatory phenotype.

### 4.2. Lipidomics in Eosinophilic and Non-Eosinophilic Asthma Phenotypes

A study by Wang et al. [107] used high-performance liquid chromatography with quadruple time-of-flight mass spectrometry (HPLC-QTOF-MS) to analyse peripheral blood lipidomics in 24 asthmatic patients and in 20 healthy controls. This study showed that, besides lipid metabolism being different between asthmatics and non-asthmatics, differences were also detected between eosinophilic and non-eosinophilic asthma. More specifically, there were significantly higher levels of phosphatidic acids and phosphatidylglycerols—PG (19:0/22:0), PG (P-18:0/18:4), PG (19:1/22:0) and PG (18:00/20:00)—in eosinophilic than in non-eosinophilic asthma.

Another recent, study that focused on sphingolipid metabolism analysed serum samples from 51 adult asthmatic patients and 9 healthy individuals by LC–MS-based target metabolomics [108]. Results also showed that the peripheral blood levels of various types of sphyngomyelin (SM), including SM 34:2, SM 38:1 and SM 40:1, were significantly decreased in asthmatics in comparison with healthy controls. More importantly, in the context of phenotypic analysis, patients with non-eosinophilic asthma had significantly lower levels of these SM subtypes than those with the eosinophilic phenotype. This finding was even more interesting since a negative correlation was found between specific SM levels and sputum IL-17 levels. Since IL-17 is a neutrophilic cytokine, this suggests that some SM may be potential biomarkers that act as protective factors in asthma and are more involved in decreasing non-eosinophilic (non-T2 high) than eosinophilic (T2 high) inflammation.

It is not clear whether the results from the previous studies by Pang et al. [91] and Guo et al. [108] are discrepant or concordant regarding sphingolipid-associated metabolism in different inflammatory asthma phenotypes, since the analysed sphingolipids were different. In any case, there is evidence that there is involvement of sphingolipids in the pathogenesis of asthma, since they play a role in cell growth, survival, inflammation and tissue remodelling. This is also suggested by a previous study in 22 adult house dust mite (HDM)-allergic asthmatics and 11 HDM-allergic rhinitis patients, who were challenged intrabronchially with a HDM (*Dermatophagoides pteronyssinus*) extract; targeted metabolomics (HPLC) was used to analyse the concentrations of various sphingolipids (sphinganine, sphinganine-1-phosphate, ceramides, sphingosine and sphingosine-1-phosphate) in peripheral blood [109]. Baseline lung function and severity of allergen-induced hyperreactivity correlated significantly with sphinganine-1-phosphate (SFA1P) and sphingosine-1-phosphate (S1P) levels, although there was no correlation with FeNO levels. Furthermore, S1P levels significantly increased in patients who developed both an early-phase and a late-phase response to allergen-specific bronchial challenge. Thus, sphingolipid-related changes may be associated with inflammatory changes induced by aeroallergen-induced bronchial hyperreactivity. Furthermore, an exploratory study in children, using untargeted metabolomics by LC–MS, showed that lower concentrations of ceramides (d:16:1/24:1 and d19:1/18:0) and sphingomyelins (and reduced concentrations of serine palmitoyl-CoA transferase, the rate-limiting enzyme in de novo sphingolipid synthesis) at the age of 6 months were associated with an increased risk of developing asthma before 3 years of age [110]. This study may be relevant because such low levels of these sphingolipids are associated with increased airway resistance, and the authors suggest that this may indicate a childhood asthma endotype with early onset and increased airway resistance, linked to reduced sphingolipid concentrations. Whether a similar pattern can also be found in adult (or elderly) asthma still needs to be ascertained.

Another study focusing on lipidomics in asthma showed that the imbalance between certain sphingolipids may be associated with different inflammatory phenotypes in patients with uncontrolled asthma [111]. This study recruited 137 adult patients with asthma and 20 healthy controls and analysed the serum levels of sphingolipids by LC–MS/MS, as well as cytokines using ELISA, besides also studying targeted gene polymorphisms to further characterise asthma inflammatory phenotypes. Asthmatics with neutrophilic asthma (determined by higher levels of CD66^+^ neutrophils) had lower lung function (lower FEV_1_%), higher Asthma Control Questionnaire (ACQ) scores (indicating worse asthma symptom control), lower Asthma Quality of Life Questionnaire (AQLQ) scores and higher sphingosine and C16:0 ceramide levels compared with those without neutrophilia. In contrast, patients with eosinophilic asthma (determined by higher levels of platelet-adherent eosinophils) had higher S1P levels compared with those without eosinophilia. This study thus showed that, in uncontrolled asthma, lipidomics may discriminate, by studying the ceramide/S1P ratio, between neutrophilic and eosinophilic asthma, with the associated implications.

A few metabolomics studies focused on lipidomics and oxidative stress, lipid peroxidation and asthma inflammatory aspects [44,55,112]. In this context, Ibrahim et al. [44] demonstrated that GC–MS analysis of exhaled breath VOCs could classify clinically relevant asthma phenotypes defined by inflammatory cells in induced sputum and asthma control, although there were no clear patterns regarding classes of metabolites (aldehydes, alkanes). This study included 35 patients with asthma and 23 healthy controls and the high accuracy of the predictive role of VOCs regarding three different asthma phenotypes (eosinophilic, neutrophilic and uncontrolled asthma) was confirmed by different models, although a validation cohort was not used.

Another study carried out by Loureiro et al. [55] aimed to analyse the relationship between oxidative stress, eosinophilic inflammation and disease severity in a sample of 57, predominantly female and atopic, non-elderly adult asthmatic patients (17 were obese, 33 had severe asthma, overwhelming majority were in GINA Steps 4 and 5), and applied comprehensive two-dimensional gas chromatography coupled to mass spectrometry (GCxGC-ToFMS) in a targeted metabolomics study of aliphatic aldehydes and alkanes in urine. This study showed that the urine of non-obese asthmatics had increased lipid metabolites that were positively associated (PLS regression) with eosinophilic inflammation, as well as with lung function and disease severity, indicating that lipidic peroxidation positively indicated an eosinophilic phenotype in these asthmatic patients. Interestingly, some biomarkers differently predicted results concerning non-metabolomics biomarkers such as FeNO or peripheral blood eosinophil numbers. In particular, the identification of metabolites such as hexane, heptane, 2,4-dimethylheptane, 2,2,4,6,6 pentamethylheptane, heptadecane, 2-methylbutanal, heptanal, 2-ethylhexanal and octanal may be associated with the presence of FeNO in the exhaled air, and the presence of heptane, octane, decane, tetradecane, pentadecane, hexadecane, octadecane, 2 methylbutanal, decanal, undecanal and hexadecanal was related to the presence of eosinophils in peripheral blood. Again, this study showed that using metabolomics may help to detect eosinophilic asthma, although the study design did not include direct comparison with non-eosinophilic asthma.

Similar results were observed by Schleich et al. [112] in a study in 521 patients with asthma, which showed that GC–MS-based metabolomics analysis of VOCs (7 potential biomarkers) in exhaled breath could differentiate among different inflammatory phenotypes. In addition, a replication study in 245 asthmatic patients confirmed four VOCs as capable of discriminating among such phenotypes. More specifically, hexane and 2-hexanone preferentially predicted eosinophilic asthma with an accuracy degree that was similar to that of FeNO and peripheral blood eosinophil numbers. In contrast, a combination of nonanal, 1-propanolol and hexane were found at higher levels in neutrophilic asthma.

Figure 3 shows the possible relationship between inflammatory asthma phenotypes and metabolic signatures.

## 5. Assessment of Metabolic Changes in Clinical Asthma Phenotypes or Endotypes

### 5.1. Metabolomics Signature in the Atopic Asthma Phenotype

Individuals were recruited from four asthma cohorts, including two adult cohorts of patients—BreathCloud (n = 407) and U-BIOPRED adults (n = 96)—in a study aimed at investigating whether eNoses could detect atopy (defined by positive skin prick tests and/or positive specific IgE levels) in pediatric and adult patients with moderate to severe asthma [113]. VOC mixtures were measured in exhaled breath using eNose technology (SpiroNose or eNose platform), and supervised data analysis was performed using three different machine learning algorithms to discriminate between atopic and non-atopic patients in training and validation sets. In addition, an unsupervised analysis was also performed using a Bayesian network, to study the relationship between eNose VOC profiles and asthma features and analyse factors that might affect it. This study showed that the characterisation of VOCs using eNoses accurately detected atopic asthma both in children and in adults, even when patients who were sensitised to non-aeroallergens (e.g., food allergens) were excluded. However, these results still need to be confirmed in other studies, namely in elderly individuals, and also using other metabolomics approaches in order to ascertain which metabolites best predict atopic asthma and allow a better understanding of the pathophysiology.

### 5.2. Metabolomics Signature in the Obese Asthma “Phenotype”

Metabolomics may also have a crucial role in allowing a better understanding of metabolic dysfunction in a broader context in asthma, by analysing the involved metabolic pathways and metabolites together with other aspects that characterise some disease phenotypes. In this setting, a recent review has dissected the role of metabolomics-driven studies of the metabolic syndrome (dyslipidemia, hypertension, insulin-resistance/type 2 diabetes mellitus) in obese asthma [88].

Overall, studies focusing on the characterisation of metabolomics signatures in exhaled breath and urine from obese patients with asthma have indicated that not only are these different from healthy controls but, importantly, they are also different from what is observed in non-obese asthmatics [61,88,114,115]. Thus, obesity-related asthma may be regarded not only as a specific phenotype but possibly also as a specific endotype, with a unique underlying pathophysiological mechanism.

Maniscalco et al. [61] studied two sets of adult volunteers: an initial, experimental set of 25 patients with obese asthma (OA), 30 patients with lean, non-obese asthma (NOA/LA) and 30 obese non-asthmatic (ONA) individuals, and a subsequent validation set involving 23 OA, 24 NOA/LA and 25 ONA. Metabolomics studies were carried out using NMR spectroscopy of exhaled breath samples and studied by discrimination of subject classes by spectral and statistical analysis, which allowed the identification of class-specific metabotypes, after metabolite profiling and metabolic pathway analysis. Obese patients had Body Mass Index (BMI) ≥ 30 Kg/m^2^ (classified as classes I, II and III obesity) and were being considered for bariatric surgery. In this study, NMR profiles showed strong regression models that allowed discrimination between OA from ONA but, very importantly, between OA and NOA/LA. Furthermore, specific biomarkers were identified which allowed this between-class separation and which are involved in methane, pyruvate and glyoxylate and dicarboxylate metabolic pathways, with many of the altered metabolites being suggestive of inflammation. More specifically, obese asthmatics had increased levels of glucose, *n*-valerate, lactate and various saturated fatty acids. Importantly, the model derived from the experimental test adequately identified 21 of the 12 OA patients and 22 of NOA/LA patients in the validation set, with high levels of accuracy, sensitivity and specificity. This study thus strongly suggested that an obese asthma metabotype may allow patient stratification reliant upon unbiased biomarkers and this may have diagnostic, therapeutic and eventually also prognostic implications.

A small, pilot study is also quite relevant in the context of a possible metabolic signature of obese asthma, and involved 11 OA and 11 NOA/LA patients and a GC–TOF–MS-based metabolomics characterisation of sputum, serum and peripheral blood mononuclear cells, with subsequent OPLS-DA and pathway topology enrichment analyses [114]. Furthermore, the researchers also studied various other immunoinflammatory parameters such as IL-1β, IL-4, IL-5, IL-6. IL-13 and TNF-α in sputum, as well as leptin, adiponectin and C-reactive protein (CRP). Overall, this study showed that there were 28 metabolites that discriminated between OA and NOA/LA. Furthermore, validation analysis identified 18 potential metabolic signatures of obese asthma. Finally, pathway topology enrichment analysis clearly showed that a broad variety of metabolic pathways were associated with obese asthma and these included cyanoamino acid metabolism, caffeine metabolism, alanine, aspartate and glutamate metabolism, phenylalanine, tyrosine and tryptophan biosynthesis and the pentose phosphate pathway in sputum, as well as glyoxylate and dicarboxylate metabolism, glycerolipid metabolism and pentose phosphate in serum. Thus, this study also confirmed obese asthma as a possible endotype with a clearly different metabotype from that observed in non-obese asthma, and which involves, among other aspects, anomalous pathways that may unbalance energy metabolism as well as oxidative stress.

### 5.3. Assessment of Metabolomics in the Steroid-Resistant Asthma “Phenotype”

Steroid-resistant asthma is generally a subtype of severe asthma and has been described in many clinical, pharmacological and immunoinflammation-focused studies in asthma. Two aspects need to be addressed in this context. The first one involves the influence that medication can have on the assessment of metabolites and asthma-related metabolic signatures, and the second one has to do with the use of metabolomics to study the response to medication.

In terms of the first issue, more persistent asthma requires treatment with ICS and, as expected, the levels of endogenous steroid molecules were found to correlate with ICS dose in a study involving a group of 54 asthmatic patients (12 mild, 20 moderate and 22 severe), and this correlation was even clearer in five patients who were on oral corticosteroids [57]. Thus, metabolomics studies on asthma and metabolic aspects of steroid metabolism will have to control for this confounding factor in order to discriminate between disease- and treatment-related effects. In any case, a study carried out in asthmatic children, using untargeted LC–MS analysis of EBC, showed that, in comparison with pre-treatment, a 3-week-long administration of ICS was not associated with significant changes in analysed metabolites [116]. Overall, metabolomics has also been shown to be a relevant tool for the study of the response (or lack of response) to corticosteroids or even the side effects of this type of medication in asthmatic patients.

Regarding the second issue, it is possible to address corticosteroid resistance, which is a hallmark of many cases of severe asthma, particularly of the neutrophilic, non-T2 type [117]. In this context, it is likely that, as was found in children [118], severe, corticosteroid-resistant asthma in adults has some specific metabolic biomarkers, although studies are lacking.

Figure 3 summarises the possible relationship between clinical asthma phenotypes and metabolic signatures, with a particular focus on discrimination between eosinophilic and neutrophilic asthma, as well as on the obesity-related asthma phenotype/endotype.

## 6. Revisiting the “Dutch Hypothesis”: Discriminating between the “Phenotypes” of Asthma and Other Chronic Obstructive Airways Diseases

It is important to ascertain whether there is any robust evidence that metabolomics can discriminate between patients with asthma and patients with other chronic obstructive diseases, namely COPD and the Asthma–COPD Overlap (ACO). This is relevant because asthma, COPD and ACO are heterogeneous, which makes differential diagnosis difficult in a significant proportion of patients. This question is even more crucial in elderly patients in whom it is more difficult to diagnose asthma in the presence of various features that are coincident with those that are present in COPD and who may also have other confounding factors, such as cardiac insufficiency.

It is possible to conceptualise asthma and COPD (and ACO) as being different expressions of the same disease (“chronic non-specific lung/airway disease”), modulated by different environmental factors, thereby eventually leading to different and/or partially overlapping forms of disease [119]. This is known as the “Dutch hypothesis”, which has been most often applied to asthma and COPD but has also more recently included ACO [120,121]. In fact, these are all “chronic obstructive airway diseases” and, in this context, asthma can be analysed as a “phenotype” of this continuum of disease expression, although this is an oversimplification since asthma, COPD and ACO have various underlying phenotypes, even with some overlap [122]. A few studies have addressed the capacity of metabolomics to discriminate between asthma, COPD and ACO, in adults.

Maniscalco et al. [52] studied 31 asthma and 44 COPD patients (ages ranging from 34 to 61 years) newly diagnosed in accordance with GINA and GOLD guidelines. These patients had relatively mild disease, and patients with cardiovascular and endocrinological comorbidities were excluded since these comorbidities might act as confounding factors. Approximately 25% of patients in each group were current smokers. NMR analysis of EBC was carried out and results were analysed in terms of orthogonal projections to latent structures discriminant analysis (OPLS-DA), a form of PCA. This study showed that it was possible to discriminate between asthma and COPD patients on the basis of OPLS-DA analysis of EBC profiles. More specifically, asthma patients had lower levels of ethanol and methanol and significantly higher levels of formate and acetone/acetoin than COPD patients. Importantly, in a second, validation cohort involving 13 asthma and 20 COPD patients, application of OPLS-DA classification confirmed the model as valid since it correctly identified 12 of 13 asthmatics and 19 of 30 COPD patients. The lower levels of methanol in asthma than in COPD may be relevant since this pro-inflammatory metabolite is increased in lung cancer patients and COPD, but not asthma, and has an increased risk of such cancer [123]. On the other hand, the higher levels of formate in asthma may be associated with specific inflammatory aspects of the disease, particularly in cases with higher bronchial reactivity to allergens [124] and with more severe disease [125]. Overall, this study by Maniscalco et al. showed that NMR metabolic profiles in EBC can be used to differentiate asthma from COPD even in smoking individuals, but this should be confirmed in patients with more severe disease and also regarding the different inflammatory phenotypes of asthma. It should be stressed that the specific subgroup analysis excluded the possibility of the asthma-related results being due to underlying atopy.

A similar discriminatory capacity between asthma and COPD was also observed in other metabolomics studies, using an electronic nose [126] and peripheral blood [127,128]. In this context, a study involving metabolomic analysis of exhaled air using an eNose in 60 asthma patients (21 with fixed airway obstruction, 39 with “classical”, reversible airway obstruction) and 40 COPD (GOLD stages II–III) showed that external validation of global breath-prints had high accuracy (88% between fixed asthma and COPD; 83% between classical asthma and COPD) for discriminating between the two diseases [126]. Importantly, the discriminatory capacity was not affected by smoking.

A similar discriminatory capacity between asthma and COPD was also observed in other metabolomics studies, using an electronic nose [126] and peripheral blood [127,128]. In this context, a study involving metabolomic analysis of exhaled air using an eNose in 60 asthma patients (21 with fixed airway obstruction, 39 with “classical”, reversible airway obstruction) and 40 COPD (GOLD stages II–III) showed that external validation of global breath-prints had high accuracy (88% between fixed asthma and COPD; 83% between classical asthma and COPD) for discriminating between the two diseases [126]. Importantly, the discriminatory capacity was not affected by smoking.

Another report analysed metabolomics-related differences in peripheral blood, between asthma and COPD [127]. Using an untargeted metabolomics approach and LC–MS, Liang et al. compared serum metabolic profiles among 17 mild persistent asthmatic patients, 17 individuals with stable COPD (2 current smokers) and 15 healthy individuals. None of the recruited individuals had significant comorbidities that might have acted as confounding factors. Overall, 19 differential metabolites were identified but the most robust results were observed with hypoxanthine, whose levels were markedly higher in asthmatic individuals than in COPD patients (and healthy controls), suggesting that purine metabolism is different between these two airway diseases.

In another study, Ghosh et al. [128] studied metabolic and immunological profiles in ACO, asthma and COPD. As previously mentioned, identification of metabolites was performed using non-targeted metabolomics with the GC–MS technique. Individuals with a diagnosis of moderate or severe asthma meeting the GINA criteria (n = 34), individuals with severe and moderate COPD according to GOLD 2014 criteria (n = 30), individuals diagnosed with ACO (n = 40) and a healthy group of smokers of the same age (n = 33) were recruited. Patients with exacerbations, viral infections and who were undergoing treatment with oral corticosteroids were excluded, to avoid these confounding factors. Importantly, for the identification of the metabolic profile, the study was carried out in two different cohorts, one for the initial, discovery phase (metabolomics and immunological profiles) and one for the validation phase (metabolomics). Metabolites such as serine, threonine, ethanolamine, glucose, D-mannose and succinic acid were downregulated in ACO compared with asthma and COPD. In contrast, the levels of 2-palmatoylglycerol and cholesterol were decreased in asthmatic individuals when compared with ACO and COPD, and the COPD group had a higher amount of these metabolites. Cholesterol is a metabolite that is intrinsically linked to inflammation and is increased in individuals with severe COPD and decreased in asthmatics. Finally, this study also showed that stearic acid expression was increased in asthma when compared with ACO and COPD, which indirectly may suggest that, similarly to linoleic acid, stearic acid may be a biomarker of involvement in the differentiation of T helper 2 (Th2) cells and T2-related inflammatory response in asthmatic individuals. Furthermore, a heatmap 2D correlation matrix in the previous study showed a significant negative correlation between serine, ethanolamine, threonine, glucose, cholesterol and succinic acid with tumor necrosis factor α (TNF-α), and non-significant negative correlations between interleukin-1β (IL-1β) and neutrophil gelatinase-associated lipocalin (NGAL) with threonine, MCP-1, YKL-40, IFN-γ and IL-6 with serine and cholesterol [128]. On the other hand, stearic acid and linoleic acid have been shown to positively correlate with TNF-α and this may be relevant since this is a pro-inflammatory cytokine that can also have increased expression not only in COPD but also in some asthmatic patients, namely in severe asthma, including refractory asthma [101] and neutrophilic severe asthma (which may be refractory or not), where, in fact, this cytokine may contribute to the neutrophilic infiltrate in the airways [129].

Thus, the metabolic profile of individuals with asthma is different from that observed in patients with other chronic obstructive respiratory disease “phenotypes” such as COPD and ACO, although further studies are warranted, involving patients with different disease phenotypes, disease severity and also, in particular, elderly patients, in whom it is even more essential to adequately diagnose these patients.

Figure 3 shows the possible relationship between asthma, ACO and COPD “phenotypes” of chronic obstructive respiratory diseases.

In order to have a general perspective of the use of different types of metabolomics approaches in asthma, Table 1 shows the main pathways and metabolites identified in such studies applied to asthma phenotypes, as well as between asthma and other chronic obstructive airways “phenotypes” such as COPD and ACO, and the techniques used in each approach.

## 7. Reproducibility and Stability of Asthma-Related Metabolic Signatures: Of Validation Cohorts, Time Stability, Age, Sex and Other Factors

The reproducibility and stability of asthma-related metabolic signatures is an issue that needs to be addressed. Such stability can be conceptualised according to various aspects: across different patient samples (external validity), over time, in different age ranges (also taking into account the elderly population), in both sexes and also in the context of comorbidities, among others.

External validity has been progressively applied in metabolomics studies in asthma, by using both an initial, primary test sample of patients and healthy controls, and a second, validation sample. Many, but not all, detected changes in primary samples were able to detect asthmatic patients in the validation sample. This has occurred not only in the discrimination of asthma versus healthy states, but also regarding severe versus milder asthma, eosinophilic versus non-eosinophilic asthma as well as asthma versus other chronic obstructive respiratory diseases. However, again, more robust evidence is necessary regarding possible links to some asthma phenotypes and this needs to be addressed in asthma endotypes other than obese-related asthma (e.g., aspirin-induced asthma or various forms of T2-high asthma).

Stability over time is also very important since it would strengthen the diagnostic and monitoring capacity of metabolomics, under similar testing conditions. Why should such stability be expected, since so many factors influence metabolic signatures? In other words, if the same cohort is studied at different timepoints, without there being significant changes in the most relevant potential confounders, it is expected that metabolomics profiles from such different timepoints will be similar? In contrast to studies regarding inflammatory or clinical phenotypes, as well as cluster analysis of such phenotypes together with relevant parameters, for which a certain degree of stability may be verified, at least for a proportion of clusters, no such studies seem to have been sufficiently carried out in metabolomics approaches to asthma. In a different setting, a small cohort study involving only healthy volunteers showed some stability in LC–MS-detected serum metabolites over time [130], but whether such temporal stability in metabolomics also applies to asthma patients still has to be adequately studied.

The reproducibility of metabolomics signatures in asthma may also be checked by comparing results between children and adult studies, since finding similar results will strengthen the possible relevance of detected changes. In fact, several studies in children and in adults have coincided in the main metabolites, at least regarding discrimination between asthma and a healthy state [32,64,88,92,94]. However, comparative studies regarding the capacity to discriminate between asthma inflammatory and/or clinical phenotypes are necessary, although the previously mentioned study by Abdel-Aziz [113], which showed the capacity of eNose metabolomics in exhaled breath to predict an atopic asthma phenotype in adults and children, is a good example of reproducibility.

In this context, it should be stated that metabolomics studies in elderly asthmatics are needed since such studies are practically non-existent since elderly patients are generally excluded from these studies, possibly because they frequently display many confounding factors, namely multimorbidity and polypharmacy.

Finally, it is also possible for gender-focused metabolomics analyses to be incorporated into future studies, since this aspect has possibly not been specifically addressed in most metabolomics reports. In other words, we cannot currently state, with a relevant degree of certainty, whether women and men have similar or different asthma metabolic signatures, although most current studies seem not to detect gender as a possible influencing factor. Nevertheless, there is some information that suggests that such gender differences in metabolic profiles may occur.

A study that used GC–MS to retrospectively analyse the metabolome in preserved umbilical cord blood from 44 children (who were 8 years old at the time of the study) showed that, among several modulatory factors, sex also had a significant influence on observed differences in metabolic pathways and their possible impact on the subsequent development of allergy [131]. This study showed that sex significantly influenced branched amino acid metabolism and vitamin B6 metabolism. In particular, the female sex was associated with significantly higher levels of leucine, isoleucine, galactitol, hydroxybutyric acid, uric acid, sucrose and mannose, and lower levels of erythritol and 2-exoisocaproic acid. Thus, this study would favour the need to differentially study metabolome signatures in men and women.

In contrast, another study applied untargeted shotgun and LC–MS methods to analyse lipidomics profiles in the lung epithelial lining fluid (obtained using induced sputum) in 41 healthy, non-smoking, adult volunteers [132]. The observed lipidome was quite diverse, including many glycerophospholipids, sphingolipids, some steroid lipids and neutral glycerolipids, and the authors were able to detect two expression phenotypes, with one of them being significantly associated with a higher BMI. However, no sex-related differences were observed. Thus, at least in healthy individuals, sex may not be a significant modulatory factor in the expression of lipidomics in the bronchial epithelium.

Finally, in a study carried out in 51 patients with asthma (17 men, 34 women), the glycerophospholipid profile was analysed in serum using an LC–MS metabolomics approach [133]. This study observed significantly different glycerophospholipid levels between men and women, with PE, PC acetal phospholipid, LPE and alkyl PE being higher in women and LPC and lysophosphatidylserine being higher in men.

Thus, it may be useful to carry out sex-specific analyses of metabolic signatures in asthma patients and determine whether any observed differences are clinically significant, namely in terms of better predicting clinical and/or inflammatory asthma phenotypes in one of the sexes. This is all the more relevant since there is epidemiological and clinical evidence showing sex-related differences in asthma, at least in part related to sex hormones, which also affect many immunophysiological parameters (reviewed by Zhang and Zein) [134].

Globally, the reproducibility and temporal stability of metabolic signatures of asthma and asthma phenotypes can only be adequately gauged by comparing the different approaches (targeted versus untargeted), the various methods that were used (e.g., NMR, various forms of MS) and the analytical parameters incorporated into validation models in each study, as well as by assessing confounding factors, the specificity of each model that was used and data obtained from different organic fluids, among other aspects.

## 8. Prognostic Value of Asthma-Related Metabolic Signatures

In order to analyse the predictive or prognostic value of asthma-related metabolic signatures, it is fundamental to consider all models used for predictive analysis, models obtained from supervised (e.g., PLS-DA, support vector machine (SVM), k-nearest neighbors algorithm (k-NN), logistic regression) and unsupervised (PCA; cluster analysis or hierarchical cluster analysis) multivariate techniques for metabolomics data analysis, since the selection of these analytic techniques influences the capacity to detect patterns, trends or potential biomarkers [135]. In particular, parameters such as classification rate, sensitivity and specificity are also important for model assessment and the relevance of their predictive potential.

Metabolomics, applied to asthma in children in several studies, has been shown to be able to predict future wheezing and exacerbations, as well as to identify different metabolites in different age groups with the same clinical profile. Thus, with the help of metabolomics, it is possible to predict an outcome for asthmatic children and eventually prevent an unfavourable exacerbation scenario [64,136,137]. However, there are comparatively fewer studies in adult asthma that have addressed this issue, in a longitudinal, cohort study. A previously mentioned, a preliminary cross-sectional study aimed at assessing urinary metabolic changes associated with asthma exacerbations was carried out by Loureiro et al. [138]. This study used targeted metabolomics involving GCxGC-ToFMS to analyse aldehydes and alkanes, and NMR to assess global changes in the major metabolites involved in the main metabolic pathways, in the urine of 10 adult asthmatic patients undergoing asthma exacerbations. It showed that asthma exacerbations were associated with increased levels of alkanes and aldehydes, which are end-products of peroxidation of unsaturated lipids and indicate underlying inflammation and high levels of oxidative stress, compared with the stable state.

Two longitudinal studies are also relevant in this context. The first one was carried out by Olopade et al. [139] and included a cross-sectional and a longitudinal component. The authors used GC-FID and showed that exhaled pentane levels increased in asthmatics undergoing an acute exacerbation, in comparison with healthy controls, and then decreased to normal levels with acute asthma treatment.

The second, longitudinal study was performed by Brinkman et al. [140] using a composite platform eNose and GC–MS in 23 currently non-smoking patients on ICS who had partially controlled, mild to persistent asthma. This study showed that a VOC model discriminated patients with baseline, stable asthma, from exacerbations, as measured by loss of control of asthma, as well as recovery. In this context, the eNose breath-print showed high accuracy, clearly higher than that of GC–MS analysis. Finally, using ANCOVA analysis, GC–MS identified two compounds (acetonitrile and bicyclo [2.2.2] octan-1-ol, 4-methyl) that positively correlated with the numbers of eosinophils (but not neutrophils) in induced sputum. Acetronitrile is a common molecule in exhaled breath and is usually associated with pathogenic bacteria [141]. In contrast, bicyclo [2.2.2] octan-1-ol, 4-methyl is akin to the 3,7,7-trimethyl-Bicyclo [4.1.0] hept-2-ene compound (also known as (+) -3-Carene) described in the study by Ibrahim et al. as being correlated with sputum eosinophils [59].

Overall, the potential role of metabolic profiles in predicting asthma exacerbations should be more thoroughly determined (also using untargeted metabolomics) in additional longitudinal studies. Furthermore, the study of other predictive features, namely in terms of prognosis (e.g., responses to treatment or lung function decline and/or non-reversibility of bronchial obstruction), is still lacking. Finally, all of these aspects should be studied in different asthma phenotypes.

## 9. Concluding Remarks and Future Challenges

Various studies have shown that metabolomics can help to distinguish between asthma and a healthy state, between severe and non-severe asthma and between asthma and other chronic obstructive respiratory diseases. In particular, and as previously mentioned in this review and also reviewed in depth by others [49,87,92,93], some metabolic pathways seem to be more consistently changed in asthma versus a healthy state.

Additionally, it also seems clear that some of the asthma inflammatory phenotypes (e.g., eosinophilic asthma) may be preferentially associated with certain metabolic signatures. However, fully demonstrable and reproducible asthma-related metabolic “phenotypes” cannot be robustly defined, with the possible exception of obesity-related asthma, which may constitute an endotype of its own and also involve a clearer metabolic phenotype with specific underlying features—that is, a “metabolic endotype”.

Thus, it is currently probably more appropriate to mention the metabolic signatures of asthma rather than actual metabolic phenotypes or endotypes. In any case, the situation will be further clarified once some of the future challenges have been dealt with. These may involve aspects such as the actual definition of clear molecular metabolic phenotypes, based on unbiased, multiple-level, integrated clustering analyses. In addition, the adequate assessment of reliable relationships between metabolic phenotypes and integrated multi-parameter phenotype clusters of asthma will be relevant in the hope that non-invasive, point-of care assessment of metabolic aspects of asthma may accurately reflect the specificities of various asthma phenotype clusters and endotypes. For this to occur, more multicentre, multinational metabolomics studies are needed, using the same techniques and similar targeted and untargeted approaches. Furthermore, the reproducibility of the metabolic signatures of asthma needs to be better defined in different settings as well as over time, in further longitudinal studies, so that the limits of variability and stability are understood for the most relevant metabolites and pathways. In addition, at least some further aspects that may affect the expression of asthma and asthma phenotype-related metabotypes should also be studied, namely nutritional aspects [142], microbiome-associated metabolic aspects [143] or air pollution parameters [144,145].

Further research is warranted and the integration of metabolomics with multi-omics and clinical–functional parameters, with subsequent artificial intelligence (AI)-driven, complex, algorithm-based analysis of “big data”, may allow a more thorough and complete analysis of integrative/global phenotype clusters of not only asthma but also within the context of chronic obstructive respiratory diseases, thereby allowing higher diagnostic yield, tailored approaches and prognostic capacity.

## Figures and Tables

**Figure 1 metabolites-11-00534-f001:**
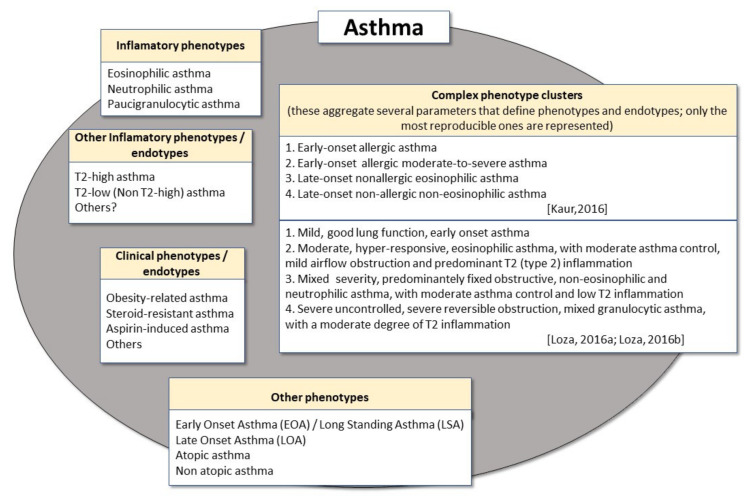
General outline of some of the most frequently analysed asthma phenotypes, endotypes and phenotype clusters in adults.

**Figure 2 metabolites-11-00534-f002:**
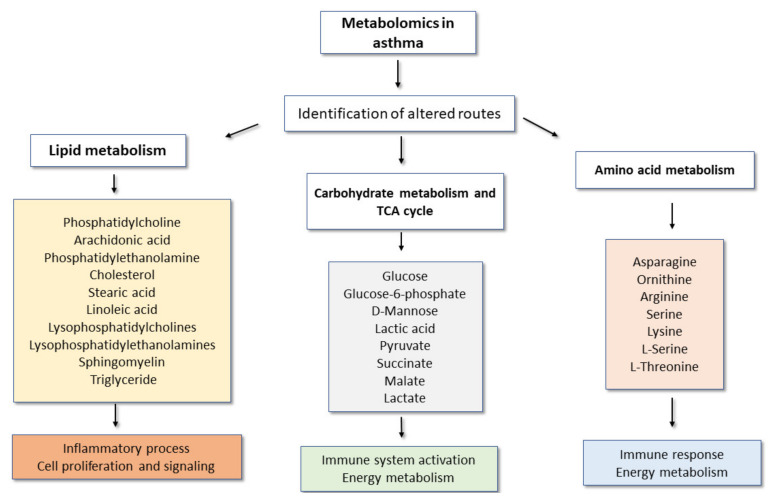
Representation of the main altered metabolic pathways and the main metabolites observed in studies of targeted and non-targeted metabolomics, as well as their relationship with some general implications for asthma. The left-hand side of the figure includes the most important pathway (lipid metabolism) and some of its metabolites, also demonstrating how these metabolites have implications for inflammatory processes, cell proliferation and signaling. In the centre, some metabolites of carbohydrate metabolism and citric acid cycle (TCA) are shown. These have been shown to have implications for activating the immune system and generating energy. On the right-hand side, some metabolites of the altered amino acid pathway are identified. These are involved in the immune response and energy metabolism.

**Figure 3 metabolites-11-00534-f003:**
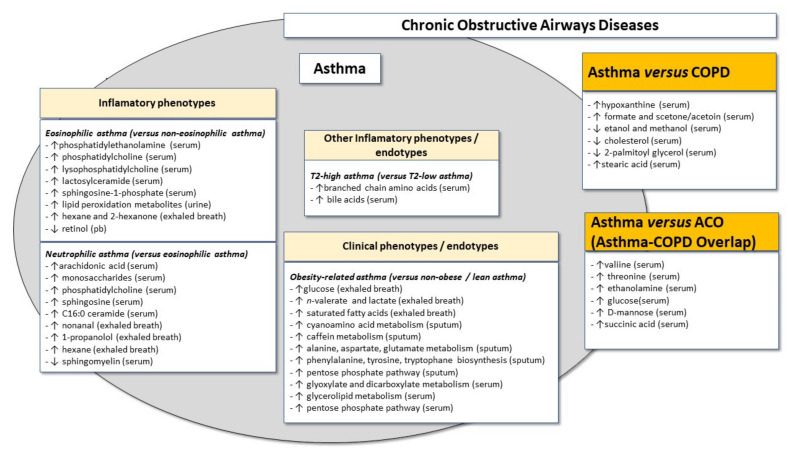
Representation of the main altered metabolic pathways and metabolites observed in metabolomics studies and their association with some of the most frequently analysed asthma phenotypes in adults. ACO—Asthma–COPD Overlap; COPD—chronic obstructive pulmonary disease; ↑-increased; ↓-decreased.

**Table 1 metabolites-11-00534-t001:** Most representative metabolomics studies in adult asthma, including relationship with inflammatory or clinical phenotypes.

Metabolomics Studies in Relation to Asthma Inflammatory Phenotypes					
**Age/Sample Size/Ref.**	**Sample Biology/** **Technique Used**	**Clinical Characteristics**	**Main Metabolites Identified**	**Main Metabolic Pathways Involved**	**Observations**
**36.4 years** **N = 20 (BA) + 10 (HC)** [54]	Peripheral blood; 3 detection platforms (UHPLC- MS/MS, optimised for basic species; UHPLC/MS/MS optimised for acidic species; GC/MS.	Patients with controlled severe asthma, patients with non-severe asthma and a healthy group (HC)	Taurine, Aspartic Acid, Glutamic Acid, Asparagine, Serine, Glutamine, Histidine, Glycine, Citrulline, Threonine, Alanine, Arginine, Tyrosine, Amino, Butyric Acid, Methionine, Valine, Tryptophan, Phenylalanine, Isoleucine, Leucine, Ornithine, Lysine 7-α-hydroxy-3-oxo-4- cholestenoate, Androsterone sulfate, Epiandrosterone sulfate, Glycerophosphorylcholine (GPC), Phosphoethanolamine, arachidonate, Oleamide Sphingosine, Glycodeoxycholate, Taurocholate, Lathosterol Adenosine 5-monophosphate	Amino acid, Carbohydrate, Lipid (Fatty acid, Sphingolipid), Bile acid, Cholesterol, Nucleotides	Biochemical differences were found between asthmatics and non-asthmatics, and also between severe and non-severe asthma; in addition, FeNo-high, possibly T2-type asthma phenotype patients had higher levels of branched amino acids and bile acids (glycholate and cholate)
**57.7 years** **N = 82 (BA) + 35 (HC)** [59]	Exhaled breath condensate (EBC); Nuclear magnetic resonance (NMR) spectroscopy	Patients with asthma-EA, NA, and a healthy control group (HC)	NMR spectral regions	Not applicable	NMR spectral regions showed potential to discriminate asthmatics from healthy controls but poorly discriminated asthma phenotypes (only NA, but not EA, could be identified)
**38 years** **N = 13 (EA) + 16 (NEA) + 15 (HC)** [91]	Peripheral blood and serum; UPLC-MS/MS	Mild and moderate asthma: 2 subgroups—EA and NEA, and a healthy control group (HC)	Glycerolphosphocholine, Monosaccharides, Phosphatidylserine (PS), Cholesterol glucuronide, Lactosylceramide, Phytosphingosine, Lysophosphatidylcholine (LPC), Retinyl ester, Retinols, Phosphatidylcholine (PC), Arachidonic acid (AA), Phosphatidylethanolamine (PE)	Glycerophospholipid, Retinol, Sphingolipid, Lipid ether, Galactose, AA, Inosite phosphate, Starch and Sucrose, Linoleic acid, Glycolysis, Gluconeogenesis	Lipid metabolism is affected in asthmatics; higher levels of monosaccharides, PC (18:1/2:0), PS (18:0/20:0) and arachidonic acid in NEA; higher levels of PC (16:0/18:1), PE (18:3/14:0), LPC (18:1) and lactosylceramide (d18:1/12:0) in EA
**48.5 years** **N = 52** [95]	Exhaled breath; Cyranose 320 eNose	Patients with persistent bronchial asthma (BA)-eosinophilic asthma (EA), various forms of non-eosinophilic asthma (NEA)—neutrophilic asthma (NA) and paucigranulocytic asthma (PGA) phenotypes	VOC breath-prints	Not applicable	Electronic nose can discriminate EA, NA and PGA inflammatory phenotypes in patients with persistent asthma in a regular clinical setting
**35.4 years** **N = 20** [96]	Bronchoalveolar lavage (BAL) Exhaled breath; eNoses	Patients with mild, allergic eosinophilic asthma (EA), who were non-smokers and not on corticosteroid therapy	eNose breath-print	Not applicable	eNose breath-prints were significantly associated with BALF eosinophil-rich inflammation
**55 years** **N = 78** [97]	Exhaled breath; eNose	Severe asthma patients-EA and NA subgroups (U-BIOPRED cohort)	Metabolomic fingerprints obtained from eNoses	Not applicable	eNose technology adequately discriminated between EA and NA (as classifed according to eosinophil and neutrophil numbers in peripheral blood, but not in induced sputum).
**Lipid metabolism**					
**46.1 years** **N = 35 (BA) + 23 (HC)** [44]	Exhaled breath VOC; GC–MS	Patients with intermittent or persistent asthma: EA and NA, and a healthy control group (HC)	Alkanes, Aldehydes	Lipid (lipid peroxidation)	Respiratory VOCs can discriminate asthmatics from non-asthmatics and identify inflammation-related disease phenotypes
**45.6 Years** **N = 57 (40 non-obese; 17 obese)** [55]	Urine; GC×GC-ToFMS	Patients with severe EA and aspirin hypersensitivity	Alkanes, Aldehydes	Lipid (lipid peroxidation)	Peroxydised lipid metabolites are increased in non-obese asthmatics and may be related to EA and disease severity.
**41 years** **N = 24 (BA) + 20 (HC)** [107]	Peripheral blood; HPLC-QTOF	Patients with asthma: 2 subgroups-EA and NEA (airway hyperresponsiveness), and a healthy control group (HC)	Fatty acyls, Glycerolipids, Glycerophospholipids, Sphingolipids, Sterol lipids and Prenol lipids	Lipid	Lipid metabolism is affected in asthmatics; significantly higher levels of phosphatidic acids and phosphatidylglycerols-PG (19:0/22:0), PG (P-18:0/18:4), PG (19:1/20:0) and PG (18:0/20:0) in EA than in NEA
**Age not indicated** **N = 51 (BA) + 9 (HC)** [108]	Serum; LC–MS	Patients with asthma: EA and NEA, early-onset asthma and late-onset asthma, and a healthy control group (HC)	Sphingomyelin (SM)	Sphingolipids	SM levels were reduced in asthma; SM (SM 34:2; SM 38:1 and SM 40:1) levels were significantly more reduced in NEA than in EA
**N = 421 (149 EA; 71 GA; 155 NA; 46 PGA)** [111]	Peripheral blood; LC–MS/MS	Patients with asthma: EA and various types of NEA—mixed granulocytic (GA), NA and PGA phenotypes	Various ceramides, Sphingosine-1-phosphate (S1P), Sphingolipids, Sphingomyelin	Lipid	Asthmatics with NA had higher sphingosine and C16:0 ceramide levels compared with those without neutrophilia; in contrast, patients with EA had higher S1P levels compared with those without eosinophilia.
**54 years** **N = 245** [112]	Exhaled breath; UHGC/MS; GCxGC-HRTOFMS	Patients with EA, NA and PGC asthma phenotypes	Alkanes, Aldehydes	Lipid (Lipid peroxidation)	VOCs discriminate between EA and NA, with hexane and 2-hexanone better identifying EA, and a combination of nonanal, 1-propanol and hexane better identifying NA
**Metabolomics studies in relation to atopic asthma phenotypes**					
**55 years** **N = 96** [113]	Exhaled breath; eNoses	Patients with mild, moderate asthma (from two adult cohorts—U-BIOPRED, BreathCloud); atopy detected by positive skin prick tests and/or allergen-specific IgE	VOC breath-prints	Not applicable	e-Nose technology can accurately and robustly differentiate between asthma patients by atopic status
**Metabolomics studies in relation to obesity-associated asthma phenotype/endotype**					
**38 years** **N = 25 (OA) + 30 (LA) + 30 (ONA)/** [61]	Exhaled breath condensate (EBC); NMR	Obese asthmatic patients (OA), lean asthmatic (LA) and obese non-asthmatic controls (ONA)	Glucose, butyrate, acetoin levels, formate, tyrosine, ethanol, ethylene glycol, methanol, acetate, saturated fatty acids, propionate levels acetoin, isovalerate, 1,2-propanediol, methanol, acetone, propionate, lactate	Carbohydrate, Lipid, Amino acid	Patients with obesity and asthma have a specific respiratory metabotype (increased levels of glucose, *n*-valerate, lactate, and various fatty acids), which is different from that of patients with obesity or asthma alone
**49 years** **N = 11 (OA) + 22 (LA)** [114]	Peripheral blood and serum Sputum supernatant; GC–TOF–MS	Obese asthmatic patients (OA), lean asthmatic (LA)	Valine, N-Methyl-DL-alanine, Uric acid, D-Glyceric acid, Asparagine 1, Beta-Glycerophosphoric acid, Benzoic acid, 3-Hydroxybutyric acid, Hydrocinnamic acid, Aspartic acid 2, Xanthine, 4-Aminobutyric acid 1, Glutaric acid, Indole-3-acetic acid, Gly-pro, D Glucoheptose, Gluconic lactone 2, L-Glutamic acid, Phytosphingosine, Shikimic acid, Beta-Glutamic acid 1, Pyrrole-2-Carboxylic, Pyrophosphate 3; 3-Aminopropionitrile 1, 3-Hydroxybutyric acid, 3-Hydroxynorvaline 2, Linolenic acid, Isoleucine	Lipid, Amino acid, Carbohydrate, Fatty acid	Metabolomics based on GC–TOF–MS discriminated between obese asthmatics and lean asthmatics
**Metabolomics studies in asthma compared with COPD and ACO**					
**48 years** **N = 31 (BA) + 44 (COPD)** [52]	Exhaled breath condensate (EBC) Proton NMR spectra	Patients with newly diagnosed asthma or COPD	Methanol, ethanol, acetone, acetaldehyde	Lipid	Asthmatics had lower levels of ethanol and methanol and significantly higher levels of formate and acetone/acetoin than COPD patients
**54 years** **N = 60 (BA)-21 (FA)) + 39 (CA) + 40 (COPD)** [126]	Exhaled air eNose	Patients with asthma (BA) with fixed airway obstruction (FA) or with classic, reversible asthma (CA); patients with COPD	Breath-prints	Not applicable	The molecular profile of exhaled breath shows high accuracy in distinguishing between FAO and COPD, as well as between CA and COPD
**60.5 years** **N = 17 (PA) + 17 (COPD) + 15 (HC)** [127]	Peripheral blood and serum; LC–MS	Patients with mild, persistent asthma (PA), COPD patients and healthy controls (HC)	Hypoxanthine; P-chlorophenylalanine; L-Glutamine; Glycerophosphocholine; Inosine; Negative ion mode (ESI-); Hypoxanthine, Succinate; Xanthine; Arachidonic Acid (peroxide free); L-Pyroglutamic acid; Indoxyl sulfate; Theophylline; L-Valine; L-Norleucine; Bilirubin; L-Leucine; Inosine; Palmitic acid; L-Phenylalanine	Lipid, Nucleic acid, Amino acid	Asthma patients have a unique serum metabolome, which can distinguish them from individuals with COPD and healthy individuals; in particular, asthmatics had significantly higher levels of hypoxanthine than COPD patients and HC
**52.7 years (Cohort 1)** **53.6 years (Cohort 2)** **N = 34 (BA)+ 30 (COPD)+ 35 (ACO)+ 33 (HC) (Cohort 1)** **N = 32 (BA) + 32 (COPD) + 40 (ACO) (Cohort 2)** [128]	Peripheral blood; GC–MS	Patients with moderate and severe asthma (BA), patients with stage II and III COPD, patients with ACO and a healthy group (HC)	L-Serine, L-threonine, Ethanolamine Glucose, D-mannose, Cholesterol, 2-palmitoylglycerol, Stearic acid, Lactic acid, Linoleic acid, Succinic acid	Carbohydrate Lipid Amino acid	2-palmatoylglycerol and cholesterol were decreased in BA when compared with ACO and COPD; in contrast, stearic acid expression was increased in BA in comparison with ACO and COPD

ACO—Asthma–COPD Overlap; BA—bronchial asthma; COPD—chronic obstructive pulmonary disease; EA—eosinophilic asthma; GC×GC-ToFMS—two-dimensional gas chromatography coupled to mass spectrometry with a high-resolution time-of-flight analyser; GC-TOF-MS—gas chromatography time of flight mass spectrometry; HPLC-QTOF—high-performance liquid chromatography with quadrupole flight time mass spectrometry; LC–MS—liquid chromatography–mass spectrometry; NA—neutrophilic asthma; NEA—non-eosinophilic asthma; PGA—paucigranulocytic asthma; UHLC/MS/MS—ultra-HPLC/tandem mass spectrometry; UPLC-MS/MS—ultra performance liquid chromatography–tandem mass spectrometry.

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
