# Peer review of "Metabolic Phenotypes in Asthmatic Adults: Relationship with Inflammatory and Clinical Phenotypes and Prognostic Implications"

_metabolites, 2021, doi:10.3390/metabo11080534_

Round 1

Reviewer 1 Report

This manuscript appears to have fairly good information content with a comprehensive review of the metabolomics associated with asthmatic adults. However, the manuscript is poorly presented due to grossly excessive use of indirect statements (throughout the entire manuscript), particularly in the first sentence (introducing most new paragraphs), with the use of adverbs, dependent clauses, and other verb forms (followed by commas), that appears more as conversational language rather than formal, conventional scientific writing. The authors could vastly improve this paper by using direct statements (especially for the first topic sentence of each paragraph), by putting the last part of the sentence first, and the first part (set off by commas) last to make the sentence direct. This should be done throughout the manuscript. Here following is a list of all paragraphs for which an indirect statement was used and could be greatly improved using direct statements and less conversational writing style:

The #s indicate the paragraph number with the sentences that needs to be rewritten (to make direct statements).

Abstract - most sentences

Paper sections (in bold) followed by paragraph numbers needing rewrites using direct statements:

Introduction - paragraphs 2, 5

Section 2. 1, 2, 3, 4, 5

3. 1, 2, 6, 7 (only 1 sentence paragraph?), 8, 9, 12, 15, 16

4. section missing entirely

5. only 1 section (don't number) 1-4, 6, 8, 9-11

6.1 1

6.2 2, 3

6.3 2, 3

7. paragraphs 1-7; paragraph 4 has only 1 disjointed sentence

8. section missing entirely

9. paragraphs 5-11

10. 3, 5, 6

11. all 4 paragraphs

Other corrections needed:

Avoid conversational phrases such as "An even more interesting study..." etc.

Table 1 - column 1 - change to: Age/ Sample size/ Ref. [put reference number in brackets]

For all Papers mentioned in the text - put reference number immediately after author's name (examples below) : [please check for this error throughout the manuscript]

Line 514 Brickman et al [90]

Line 533 Fens et al [87]

Line 547 Plaza et al [98]

Line 558 Schleich et al [89]

References 2 and 11 lack second-line indents

Remove Acknowledgments (if none are to be indicated)

Author Response

REVIEWER 1

Comments and Suggestions for Authors

This manuscript appears to have fairly good information content with a comprehensive review of the metabolomics associated with asthmatic adults. However, the manuscript is poorly presented due to grossly excessive use of indirect statements (throughout the entire manuscript), particularly in the first sentence (introducing most new paragraphs), with the use of adverbs, dependent clauses, and other verb forms (followed by commas), that appears more as conversational language rather than formal, conventional scientific writing. The authors could vastly improve this paper by using direct statements (especially for the first topic sentence of each paragraph), by putting the last part of the sentence first, and the first part (set off by commas) last to make the sentence direct. This should be done throughout the manuscript. Here following is a list of all paragraphs for which an indirect statement was used and could be greatly improved using direct statements and less conversational writing style:

The #s indicate the paragraph number with the sentences that needs to be rewritten (to make direct statements).

Reply: Dear reviewer, we really thank you for your careful revision of our manuscript; your suggestions have definitely allowed us to improve its quality and we have followed as much as possible or feasible, to make it more direct, without affecting the personal narrative style.

Abstract - most sentences

Reply: We have endeavoured to change it as suggested

Paper sections (in bold) followed by paragraph numbers needing rewrites using direct statements:

Introduction - paragraphs 2, 5

Reply: Corrected

Section 2. 1, 2, 3, 4, 5

Reply: Corrected

  1. 1, 2, 6, 7 (only 1 sentence paragraph?), 8, 9, 12, 15, 16

Reply: Corrected. Paragraph 7 contains a single sentence because it focuses on ethane levels, and this focus is different from that in the previous (aminoacids) and subsequent (carbohydrates) paragraphs

  1. section missing entirely

Reply: This has been corrected; section 5 is now current section 4

  1. only 1 section (don't number) 1-4, 6, 8, 9-11

Reply: Corrected as adequately as it was feasible. Sub-sections have been inserted.

6.1 1

6.2 2, 3

6.3 2, 3

Reply: These sentences were corrected where we believed they could be improved.

  1. paragraphs 1-7; paragraph 4 has only 1 disjointed sentence

Reply: We have corrected this issue.

  1. section missing entirely

Reply: Corrected; this is now section 7

  1. paragraphs 5-11
  2. 3, 5, 6
  3. all 4 paragraphs

Reply: These sentences were corrected where we believed they could be improved.

Other corrections needed:

Avoid conversational phrases such as "An even more interesting study..." etc.

Reply: Dear reviewer, we haver e-checked the text and removed this type of statements as much as we could.

Table 1 - column 1 - change to: Age/ Sample size/ Ref. [put reference number in brackets]

Reply: We have done as suggested

For all Papers mentioned in the text - put reference number immediately after author's name (examples below) : [please check for this error throughout the manuscript]

Line 514 Brickman et al [90]

Line 533 Fens et al [87]

Line 547 Plaza et al [98]

Line 558 Schleich et al [89]

Reply: We have corrected all in-text references to papers, when authors’ names were mentioned, as suggested

References 2 and 11 lack second-line indents

Reply: Corrected

Remove Acknowledgments (if none are to be indicated)

Reply: Done as suggested

Reviewer 2 Report

With interest, I read the manuscript metabolites-1276040. The Authors did a great job and invested lots of work into this manuscript. The main text is very interesting, figures comprehensive, clear, and elegant, and the table is impressive. I have some additional comments for consideration.

Comments:

  1. Starting from the title, I would change to “Metabolomic phenotypes in asthmatic adults: relationship to inflammatory and clinical phenotypes, and prognostic implications”.
  2. I suggest to refer somewhere in your article to the paper PMID: 33339279, a recent review published in Metabolites and describing some other aspects of allergy metabolomics, partly complementary to the content of your article.
  3. I also suggest incorporating this original article recently published in Metabolites: PMID: 33003349. It refers to asthma breathomics in the context of inhaled corticosteroids.
  4. While discussing obesity-associated asthma, please refer to PMID: 30057383 as it refers to some aspects not covered by the article you are discussing.
  5. The nutritional links of asthma metabolomics could be addressed a bit more, maybe in 2-3 sentences, especially in the context of epigenetics (linked to metabolism also independently to nutrition). Please, refer to PMID: 33668787.
  6. You could consider creating a graphical abstract.

Author Response

REPLIES TO REVIEWER 2

Comments and Suggestions for Authors

With interest, I read the manuscript metabolites-1276040. The Authors did a great job and invested lots of work into this manuscript. The main text is very interesting, figures comprehensive, clear, and elegant, and the table is impressive. I have some additional comments for consideration.

Comments:

  1. Starting from the title, I would change to “Metabolomic phenotypes in asthmatic adults: relationship to inflammatory and clinical phenotypes, and prognostic implications”.

Reply: We agree that the suggested title may be better but, of we are to change from “metabolomics”, it may be even better to switch to “Metabolic”. We have made this correction.

  1. I suggest to refer somewhere in your article to the paper PMID: 33339279, a recent review published in Metabolites and describing some other aspects of allergy metabolomics, partly complementary to the content of your article.

Reply: We thank the reviewer for this suggestion; we have added this article as current Reference 49 and mentioned in line 139.

  1. I also suggest incorporating this original article recently published in Metabolites: PMID: 33003349. It refers to asthma breathomics in the context of inhaled corticosteroids.

Reply: Thank you for this suggestion. We have incorporated this article as Reference 104 and included a sentence in currnent Section 5.3 (lines 682-685).

  1. While discussing obesity-associated asthma, please refer to PMID: 30057383 as it refers to some aspects not covered by the article you are discussing.

Reply: Thank you for the suggestion. We have included this article as Reference 103 (line 625), but could not include any sentence since the editor asked us to shorten the manuscript.

  1. The nutritional links of asthma metabolomics could be addressed a bit more, maybe in 2-3 sentences, especially in the context of epigenetics (linked to metabolism also independently to nutrition). Please, refer to PMID: 33668787.

Reply: Dear reviewer, thank you for your suggestion. We agree that epigenetics are also relevant in terms of metabolismo but discussing this issue would possibly make us veer off a little bit from our main focus (metabolomics in asthma phenotypes), and we really have to follow the Editor’s suggestion to shorten the manuscript and keep the focus on phenotypes. Thus, we have only briefly included this issue in the final comments (line 980) and added the suggested reference (Reference 131).

  1. You could consider creating a graphical abstract.

Reply: We really thank the reviewer for this suggestion and have added a graphical abstract.

Reviewer 3 Report

The authors attempted a conceptual approach to the metabolomics signatures and asthma phenotypes issue in patients with asthma-related diseases, shedding various lights on inflammatory, clinical, and prognostic implications from the metabolomics perspective.

This work has provided a stepping stone for a personalized evaluation of asthma-related diseases with various faces. Thus, it is expected to provide a theoretical foundation for precision medical approaches through artificial intelligence and big data in the future.

It would be better to address some concerns for scientific soundness before the acceptance of the manuscript.

  1. Considering the paper's content, it would be better not to confine the title to adults, and it would be appropriate to express it covering not only asthma but also COPD and ACO.
  1. This study is a paper described by several authors in the form of a narrative review and has found several formal errors. First of all, items corresponding to numbers 4 and 8 are missing. Second, 5.1 is the only detail item presented in item 5, but it is recommended to divide the details further because it is described too long.
  1. Multiple typos and grammatical errors are found throughout the paper. In addition, it is recommended that the acronym be described in parentheses only once in the beginning and then consistently as an abbreviation.

Author Response

REPLIES TO REVIEWER 3

Comments and Suggestions for Authors

The authors attempted a conceptual approach to the metabolomics signatures and asthma phenotypes issue in patients with asthma-related diseases, shedding various lights on inflammatory, clinical, and prognostic implications from the metabolomics perspective.

This work has provided a stepping stone for a personalized evaluation of asthma-related diseases with various faces. Thus, it is expected to provide a theoretical foundation for precision medical approaches through artificial intelligence and big data in the future.

It would be better to address some concerns for scientific soundness before the acceptance of the manuscript.

  1. Considering the paper's content, it would be better not to confine the title to adults, and it would be appropriate to express it covering not only asthma but also COPD and ACO.

Reply: Dear reviewer, we thank you for your insightful comment; however, we believe that it will be more adequate to keep the focus on adults, since there have been some recente reviews on various aspects of metabolomics in children asthma; in addition, the main focus is asthma; COPD and ACO only feature in this review as a comparator since they are a form (just like asthma) of chronic obstructive airways disease; thus, all data about COPD and ACO that feature in this review only concern a minor section of the review and only feature as a way to compare asthma with them, regarding metabolomics; we believe that a future, more detailed review of metabolomics in all chronic obstructive airways disease is warranted.

  1. This study is a paper described by several authors in the form of a narrative review and has found several formal errors. First of all, items corresponding to numbers 4 and 8 are missing. Second, 5.1 is the only detail item presented in item 5, but it is recommended to divide the details further because it is described too long.

Reply: Dear reviewer, we thank you for these important comments. We have now removed previously empty sections 4 and 8 and renumbered all the sections.

We have also subdivided section 5.1 (this is now Section 4 and its subdivisions).

  1. Multiple typos and grammatical errors are found throughout the paper. In addition, it is recommended that the acronym be described in parentheses only once in the beginning and then consistently as an abbreviation.

Reply: Dear reviewer, we thank you for these comments, which are also in line with similar comments from the Editor. We have carefully re-checked the manuscript and believe that we have eliminated all typos and gramatical errors. In addition, we have made sure that acronyms are defined when a term first appears in the text.

Reviewer 4 Report

The review of Adalberto  Santos and Coll. is an interesting manuscript which summarizes the most important  research on metabolomics, focused on the discrimination between asthma and healthy state and the metabolic signatures which may identify some asthma phenotypes or endotypes. As Authors underline, metabolomics is a promising  tool, which may prove useful in point-of-care application.

Minor point

I suggest Authors to mention the influence of air pollution on metabolic pathways, as air pollution may be an influent factor which lead to a specific asthma phenotype.

Using a global untargeted metabolomic approach, Nassan and Coll identified several significant metabolites and metabolic pathways associated with long-term exposure to PM2.5, NO2 and temperature (Metabolomic signatures of the long-term exposure to air pollution and temperature. Nassan FL et al Environmental Health 2021) and perturbation of metabolic pathways has been shown to mediate the association of air pollutants with asthma and cardiovascular diseases(Jeong A et al. Environment International 2018)

Author Response

REPLIES TO REVIEWER 4

Comments and Suggestions for Authors

The review of Adalberto  Santos and Coll. is an interesting manuscript which summarizes the most important  research on metabolomics, focused on the discrimination between asthma and healthy state and the metabolic signatures which may identify some asthma phenotypes or endotypes. As Authors underline, metabolomics is a promising  tool, which may prove useful in point-of-care application.

Minor point

I suggest Authors to mention the influence of air pollution on metabolic pathways, as air pollution may be an influent factor which lead to a specific asthma phenotype.

Using a global untargeted metabolomic approach, Nassan and Coll identified several significant metabolites and metabolic pathways associated with long-term exposure to PM2.5, NO2 and temperature (Metabolomic signatures of the long-term exposure to air pollution and temperature. Nassan FL et al Environmental Health 2021) and perturbation of metabolic pathways has been shown to mediate the association of air pollutants with asthma and cardiovascular diseases(Jeong A et al. Environment International 2018)

Reply: Dear reviewer, many thanks for your comments and suggestion. We fully agree that air pollution may affect metabolic pathways and this may have an impact on asthma. However, since the Editor asked us to keep a main focus (and which is, for this review, metabolomics in asthma phenotypes), we have only briefly added information regarding this issue in the conclusions section (line 981) and added the two references (References 133 and 134).

Round 2

Reviewer 1 Report

The authors have made a strong attempt to write a comprehensive review to address the detection of asthma and related diseases through breath analysis. There are a few significant omissions and sentence structure revisions needed to improve the quality and clarity of the manuscript as follows.

Entire manuscript revisions:

Writing style could be significant improved and more professional through the avoidance of conversational language and excessive use of indirect statements in topic sentences (i.e., the first sentence in each new paragraph). The use of indirect statements in topic sentences (introduced using either prepositions, dependent clauses, or adverbs (all followed by a comma) results in a weak statement that does not effectively express the main topic of the paragraph. This can be easily corrected in topic sentences by putting the subject of the sentence at the beginning and moving the first part to the end of the sentence to make a much stronger direct statement. Please correct this throughout the manuscript.

Introduction

Reference deficiencies (and omissions of key concepts)

The Introduction omits some key applications and references of e-nose technologies that provide more rapid early detection capabilities than do analytical chemistry-based methods. Please add the following key concepts (related to these points) along with the associated suggested references that introduce these concepts:

  1. Disease-specific VOC-biomarkers to Asthma and associated lung diseases:

Some general recent references

Metabolites 5(1), 140-163 (2015)

Curr. Metabolomics 5, 90-101 (2017)

Sensors 11, 1105-1176 (2011)

Asthma

Acta Physiol. (Oxf.) 189, 87–98 (2007)

Am. J. Respir. Crit. Care Med. 169, 473–478 (2004)

COPD

Thorax 54, 572–575 (1999)

Cystic Fibrosis (CF)

Eur. Respir. J.  17, 1201–1207 (2001)

Proc. Natl. Acad. Sci. USA 102, 15762–15767 (2005)

Eur. Respir. J. 27, 929–936 (2006)

Tuberculosis (TB)

Tuberculosis 89, 263–266 (2009)

Upper Respiratory Tract Infections (URTI)

Microbiology 158, 3044–3053 (2012)

         2. Some VOC-biomarkers are indicative of different diseases (such as nitric oxide, indicating asthma and COPD diseases)

Acta Physiol. (Oxf.) 189, 87–98 (2007)

Am. J. Respir. Crit. Care Med. 169, 473–478 (2004)

 Eur. Respir. J. 17, 1201–1207. (2001)

3. Uses of E-nose devices to detect disease-specific unique mixtures of VOC-metabolites (disease biomarkers); [reference sources]

Advantages of e-noses: faster, cheaper, noninvasive, and much earlier detection than conventional analytical chemistry diagnostic methods. More recent dual-technology e-noses have chemical analysis capabilities to identify VOC-biomarkers of disease in the human breath. Also mention the use of application-specific database libraries of VOC-biomarkers for early disease detection.

Chemosensors 6, 45 (2018)

J. Med. Surg. Pathol. 3, 4 (2018)

Biosensors 10, 73 (2020)

Merge the 3rd and 4th Introduction paragraphs (on Metabolomics) as a single paragraph

Author Response

Replies to comments from Reviewer 1

Comments and Suggestions for Authors

The authors have made a strong attempt to write a comprehensive review to address the detection of asthma and related diseases through breath analysis. There are a few significant omissions and sentence structure revisions needed to improve the quality and clarity of the manuscript as follows.

1) Entire manuscript revisions:

Writing style could be significant improved and more professional through the avoidance of conversational language and excessive use of indirect statements in topic sentences (i.e., the first sentence in each new paragraph). The use of indirect statements in topic sentences (introduced using either prepositions, dependent clauses, or adverbs (all followed by a comma) results in a weak statement that does not effectively express the main topic of the paragraph. This can be easily corrected in topic sentences by putting the subject of the sentence at the beginning and moving the first part to the end of the sentence to make a much stronger direct statement. Please correct this throughout the manuscript.

REPLY: Dear reviewer, we have again tried to improve this aspect and believe that the text reads better, now. Many thanks for your helpful comments regarding style.

2) Introduction

2.1.) Reference deficiencies (and omissions of key concepts)

The Introduction omits some key applications and references of e-nose technologies that provide more rapid early detection capabilities than do analytical chemistry-based methods. Please add the following key concepts (related to these points) along with the associated suggested references that introduce these concepts:

REPLY: Dear authors, your suggestion is quite pertinente and useful and we really thank you for that. Having re-read the review we fully agree that eNose concepts should be briefly introduced. We do hope that we have made this in a succinct but clear and helpful way.

References below have been added as suggested.

  1. Disease-specific VOC-biomarkers to Asthma and associated lung diseases:

Some general recent references

- Wilson AD. Advances in electronic-nose technologies for the detection of volatile biomarker metabolites in the human breath. Metabolites 2015, 5, 140-163

- Wilson AD. Biomarker metabolite signatures pave the way for electronic-nose applications in early clinical disease diagnoses. Curr. Metabolomics 2017, 5, 90-101. doi: 10.2174/2213235X04666160728161251.

- Wilson AD, Baietto M. Advances in electronic-nose technologies developed for biomedical applications. Sensors 2011, 11, 1105-1176. Doi: 10.3390/s110101105.

Asthma

- Lärstad MAE, Torén K, Bake B, Olin A-C. Determination of ethane, pentabe and isoprene in exhaled air – effects of breath-holding, flow rate and purified air. Acta Physiol. (Oxf.) 2007, 189, 87–98. doi: 10.1111/j.1748-1716.2006.01624.x.

- Smith AD, Cowan JO, Filsell S, McLachlan C, Monti-Sheehan G, Jackson P, et al. Diagnosing asthma: comparisons between exhaled nitric oxide measurements and conventional tests. Am. J. Respir. Crit. Care Med. 2004, 169, 473–478. doi: 10.1164/rccm.200310-1376OC.

COPD

- Corradi M, Majori M, Cacciani GC, Consigli GF, de’Munari E, Pesci A. Increased exhaled nitric oxide in patients with stable chronic obstructive pulmonar disease. Thorax 1999, 54, 572–575. doi: 10.1136/thx.54.7.572.

- Binson VA, Subramoniam M, Mathew L. Discrimination of COPD and lung cancer from controls through breath analysis using a self-developed e-nose. J Breath Res. 2021 Jul 9. doi: 10.1088/1752-7163/ac1326.

- Ratiu IA, Ligor T, Bocos-Bintintan V, Mayhew CA, Buszewski B. Volatile Organic Compounds in exhaled breath as fingerprints of kung cancer, asthma and COPD. J Clin Med. 2020, 10, 32. doi: 10.3390/jcm10010032. 

Cystic Fibrosis (CF)

- Balint B, Kharitonov SA, Hanazawa T, Donnelly LE, Shah PL, et al. Increased nitrotyrosine in exhaled breath condensate in cystic fibrosis. Eur. Respir. J. 2001, 17, 1201–1207.

- Kamboures MA, Blake DR, Cooper DM, Newcomb RL, Barker M, Larson JK, et al. Breath sulfides and pulmonar function in cystic finrosis. Proc. Natl. Acad. Sci. USA 2005, 102, 15762–15767. doi: 10.1073/pnas.0507263102.

- Barker M, Hengst M, Schmid J, Buers H-J, Mittermaier B, Klemp D, et al. Volatile organic compounds in the exhaled breath of young patients with cystic fibrosis. Eur. Respir. J. 2006, 27, 929–936. doi: 10.1183/09031936.06.0085105.

Tuberculosis (TB)

- Syhre M, Manning L, Phuanukoonnon S, Harino P, Chambers ST. The scent of Mycobacterium tuberculsos – Part II breath. Tuberculosis 2009, 89, 263–266. doi: 10.1016/j.tube.2009.04.003.

Upper Respiratory Tract Infections (URTI)

- Flipiak W, Sponring A, Baur MM, Ager C, Filipiak A, Wiesenhofer H, et al. Characterization of volatile metabolites taken up by or released from Streptococcus pneumoniae and Haemophilus influenzae by using GC-MS. Microbiology (Reading) 2012, 158, 3044–3053. doi: 10.1099/mic.0.062687-0.

  1. Some VOC-biomarkers are indicative of different diseases (such as nitric oxide, indicating asthma and COPD diseases)

- Lärstad MAE, Torén K, Bake B, Olin A-C. Determination of ethane, pentabe and isoprene in exhaled air – effects of breath-holding, flow rate and purified air. Acta Physiol. (Oxf.) 2007, 189, 87–98. doi: 10.1111/j.1748-1716.2006.01624.x.

- Smith AD, Cowan JO, Filsell S, McLachlan C, Monti-Sheehan G, Jackson P, et al. Diagnosing asthma: comparisons between exhaled nitric oxide measurements and conventional tests. Am. J. Respir. Crit. Care Med. 2004, 169, 473–478. doi: 10.1164/rccm.200310-1376OC.

- Balint B, Kharitonov SA, Hanazawa T, Donnelly LE, Shah PL, et al. Increased nitrotyrosine in exhaled breath condensate in cystic fibrosis. Eur. Respir. J. 2001, 17, 1201–1207.

  1. Uses of E-nose devices to detect disease-specific unique mixtures of VOC-metabolites (disease biomarkers); [reference sources]

Advantages of e-noses: faster, cheaper, noninvasive, and much earlier detection than conventional analytical chemistry diagnostic methods. More recent dual-technology e-noses have chemical analysis capabilities to identify VOC-biomarkers of disease in the human breath. Also mention the use of application-specific database libraries of VOC-biomarkers for early disease detection.

- Wilson, A.D. Applications of electronic-nose technologies for noninvasive early detection of plant, animal and human diseases. Chemosensors 2018, 6, 45. doi: 10.3390/chemosensors6040045.

- Wilson, AD. Developing electronic-nose technologies for clinical practice. J. Med. Surg. Pathol. 2018, 3, 4. doi: 10.4172/2472-4971.1000168.

- Wilson, AD. Noninvasive early disease diagnosis by electronic-nose and related VOC-detection devices. Biosensors 2020, 10, 73. 10.3390/bios10070073.

2.2. Merge the 3rd and 4th Introduction paragraphs (on Metabolomics) as a single paragraph

REPLY: We have done as requested.

GLOBAL REPLY:

Dear author, we must really thank you for your thorough, insightful and helpful feedback regarding this review. Your comments and suggestions have definitely improved the quality of the manuscript.

Round 3

Reviewer 1 Report

The authors have made a good effort to include additional pertinent references in the Introduction that were deficient relative to the use of electronic-nose devices for early detection of respiratory diseases. There are a couple of suggested revisions of newly added text to improve the clarity of statements as follows.

Page 3, Line 111: reword (remove beginning preposition)

Electronic nose (eNose) devices can be used for global metabolite characterization, by providing breath-prints mostly involving Volatile Organic Compounds (VOCs).

Page 3, Line 116: reword

Electronic noses can detect changes in VOCs in asthma [30, 31, 32], COPD [33-35], as well as in various other respiratory diseases [36-39]. 

Page 3, Line 118; please reword as follows

Dual-technology eNoses are similar to conventional chemical identification approaches in having chemical analysis capabilities that allow them to identify VOCs as disease-specific biomarkers [30-35].

Author Response

There are a couple of suggested revisions of newly added text to improve the clarity of statements as follows.

Reply: Thank you for once again carefully analysing our manuscript and making suggestions to strengthen it .

Page 3, Line 111: reword (remove beginning preposition)

Electronic nose (eNose) devices can be used for global metabolite characterization, by providing breath-prints mostly involving Volatile Organic Compounds (VOCs).

Reply: We have modified this sentence in order to make it clearer. It now reads "Electronic nose (eNose) devices can be used for global metabolite characterization, by detecting complex mixtures of Volatile Organic Compounds (VOCs) in exhaled breath, and providing associated breath-prints of such mixtures. "

Page 3, Line 116: reword

Electronic noses can detect changes in VOCs in asthma [30, 31, 32], COPD [33-35], as well as in various other respiratory diseases [36-39]. 

Reply: We have changed this sentence, as suggested. It now reads ", eNoses can detect changes in VOCs mixtures in asthma [30, 31, 32], COPD [33-35], as well as in various other respiratory diseases, namely cystic fibrosis or tuberculosis [36-39]. "

Page 3, Line 118; please reword as follows

Dual-technology eNoses are similar to conventional chemical identification approaches in having chemical analysis capabilities that allow them to identify VOCs as disease-specific biomarkers [30-35].

Reply: We reworded as requested.